# Electric-field control of skyrmions in multiferroic heterostructure via magnetoelectric coupling

You Ba[1,2,8], Shihao Zhuang[3,8], Yike Zhang[1,2,8], Yutong Wang[1,2], Yang Gao[4], Hengan Zhou[1,2], Mingfeng Chen [5], Weideng Sun[1,2], Quan Liu[1,2], Guozhi Chai [4], Jing Ma [5], Ying Zhang [6], Huanfang Tian[6], Haifeng Du[7], Wanjun Jiang[1,2], Cewen Nan[5], Jia-Mian Hu [3✉] & Yonggang Zhao [1,2✉]

Room-temperature skyrmions in magnetic multilayers are considered to be promising candidates for the next-generation spintronic devices. Several approaches have been developed to control skyrmions, but they either cause significant heat dissipation or require ultrahigh electric fields near the breakdown threshold. Here, we demonstrate electric-field control of skyrmions through strain-mediated magnetoelectric coupling in ferromagnetic/ferroelectric multiferroic heterostructures. We show the process of non-volatile creation of multiple skyrmions, reversible deformation and annihilation of a single skyrmion by performing magnetic force microscopy with in situ electric fields. Strain-induced changes in perpendicular magnetic anisotropy and interfacial Dzyaloshinskii–Moriya interaction strength are characterized experimentally. These experimental results, together with micromagnetic simulations, demonstrate that strain-mediated magnetoelectric coupling (via strain-induced changes in both the perpendicular magnetic anisotropy and interfacial Dzyaloshinskii–Moriya interaction is responsible for the observed electric-field control of skyrmions. Our work provides a platform to investigate electric-field control of skyrmions in multiferroic heterostructures and paves the way towards more energy-efficient skyrmion-based spintronics.

---

[1] Department of Physics, State Key Laboratory of Low-Dimensional Quantum Physics, Tsinghua University, Beijing 100084, China. [2] Frontier Science Center for Quantum Information, Tsinghua University, Beijing 100084, China. [3] Department of Materials Science and Engineering, University of Wisconsin-Madison, Madison, WI 53706, USA. [4] Key Laboratory for Magnetism and Magnetic Materials of the Ministry of Education, Lanzhou University, Lanzhou 730000, China. [5] School of Materials Science and Engineering, State Key Laboratory of New Ceramics and Fine Processing, Tsinghua University, Beijing 100084, China. [6] Beijing National Laboratory for Condensed Matter Physics, Institute of Physics, Chinese Academy of Sciences, Beijing 100190, China. [7] Anhui Province Key Laboratory of Condensed Matter Physics at Extreme Conditions, High Magnetic Field Laboratory of Chinese Academy of Sciences, University of Science and Technology of China, Hefei 230031, China. [8] These authors contributed equally: You Ba, Shihao Zhuang, Yike Zhang. ✉email: jhu238@wisc.edu; ygzhao@tsinghua.edu.cn

Skyrmions are topologically protected particle-like spin textures and have potential applications in information storage, such as, skyrmion-based racetrack memory, due to their small size and high mobility[1,2]. Typically, skyrmions are stabilized by the Dzyaloshinskii–Moriya interaction (DMI) originated from the spin–orbit coupling in a noncentrosymmetric chiral helimagnet[2–4]. In B20-type compounds[5], skyrmions exist in a narrow temperature and magnetic field phase diagram, usually far away from room temperature[6,7], which is unfavorable for applications. By contrast, the asymmetric multilayer stacks consisting of heavy metal (HM) and ferromagnetic (FM) can stabilize Néel-type skyrmions at room temperature and enable fast current-driven skyrmion motion[8,9], due to the interfacial DMI induced by the broken interfacial inversion symmetry and the large spin–orbit coupling of HM[10–12]. The multilayer structure is particularly appealing, because of the variety of materials and the enhanced interfacial DMI of HM/FM interfaces[13]. Furthermore, due to the compatibility with standard semiconductor techniques[14], room-temperature skyrmions in multilayers are promising candidates for the next-generation spintronic devices with high-density, low-power consumption and nonvolatility[2].

The ability to control (that is, create[15], delete[16] and move[17]) skyrmions by external stimuli are essential for applications. Electrical stimuli, such as spin-polarized current[18] and electric-field gating[15], are most industrially compatible. However, electric currents dissipate a significant amount of heat, while the ultra-high electric fields currently used in electric-field gating is near the breakdown threshold value of the dielectric layer[19]. From a fundamental perspective, due to the short charge-screening length of the metallic ferromagnet, electric-field gating mainly affects the FM/dielectric interface[19–22], with little influence on the HM/FM interface including the interfacial DMI.

Strain-mediated electric-field control of magnetism in multiferroic heterostructure consisting of FM and ferroelectric layers has been widely studied[23–25]. In contrast to electric-field gating, electric-field-induced strains in the ferroelectric can be transferred to the FM layer over hundreds of nanometers[26] or more,

which should lead to a more effective skyrmion manipulation by modulating both the magnetic anisotropy and the interfacial DMI of the HM/FM interface. Notably, electric-field control of the interfacial DMI of the HM/FM interface has so far only been observed through ionic-liquid-gating-induced carrier density change[27], and the ionic-liquid may damage the sample. Instead, a nondestructive HM/FM interfacial DMI modulation could be achieved via the strain-mediated electric-field control in all-solid-state heterostructures/devices. Strain-mediated electric-field-induced creation[28], deletion,[29] and motion[30] of skyrmions have been computationally demonstrated.

In this work, we demonstrate strain-mediated electric-field control of skyrmions through magnetoelectric coupling in multiferroic heterostructure composed of [Pt/Co/Ta] × 5 magnetic multilayer and ferroelectric $Pb(Mg_{1/3}Nb_{2/3})_{0.7}Ti_{0.3}O_3$ (PMN-PT). Through performing magnetic force microscopy (MFM) measurements with in situ electric fields, we demonstrate creation, deformation, and annihilation of skyrmions. Electric-field control of the interfacial DMI of the HM/FM interface and perpendicular magnetic anisotropy (PMA) are also demonstrated experimentally. Micromagnetic simulations are performed for understanding the mechanisms of skyrmion manipulation. Our findings will stimulate more research for electric-field control of skyrmions in multiferroic heterostructures, with application to skyrmion-based low-power spintronic devices.

## Results

**Multilayer structure.** Ta(4.7 nm)/[Pt(4 nm)/Co(1.6 nm)/Ta(1.9 nm)] × 5 (hereafter Pt/Co/Ta) multilayers were deposited by using ultrahigh vacuum magnetron sputtering on PMN-PT(001) single-crystal substrate (as shown in Fig. 1a), and separately on a 30-nm-thick $Si_3N_4$ membrane for Lorentz transmission electron microscopy (L-TEM) observation. The stack structure and setup for MFM measurement with in situ electric fields are schematically shown in Supplementary Fig. 1. The bottom Ta layer was used to flat the surface of substrate and promote the interface

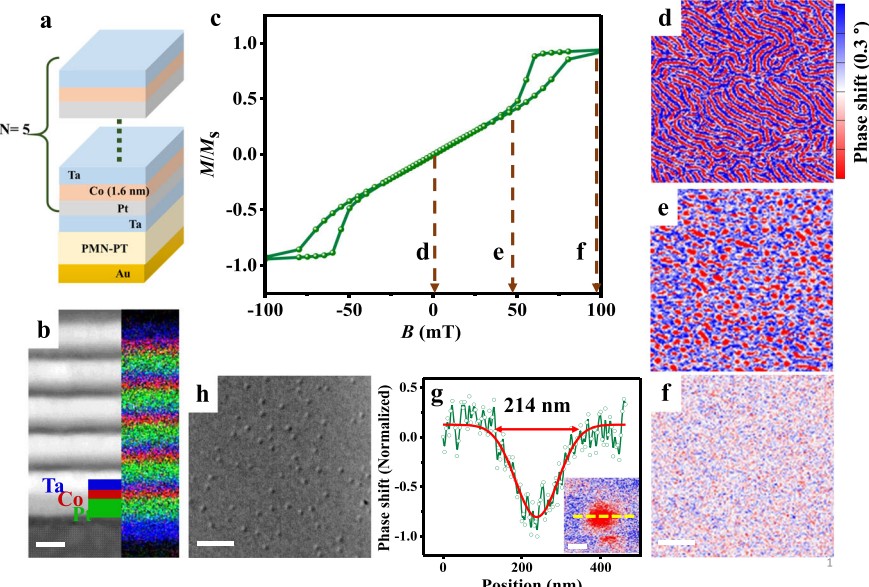

**Fig. 1 Structure and skyrmion characterizations. a** Schematic of the sample configuration. **b** Cross-sectional HAADF-STEM image of the multilayer and corresponding EDX mapping for element distributions of Pt (green), Co (red) and Ta (blue) with a scale bar of 5 nm. **c** Out-of-plane magnetic hysteresis. **d–f** Magnetic-domain evolutions with increasing magnetic field with the same scale bar of 1 μm. **g** Line-cut profile along the yellow dot line in the inset with Gaussian fitting (red curve), the inset is magnified individual skyrmion with a scale bar of 100 nm. **h** Representative L-TEM image of skyrmions with a scale bar of 1 μm.

quality[14], which enables a large PMA in the multilayer and a strong interfacial DMI at the Pt/Co interface[31]. Pt/Co and Ta/Co interfaces generate strong interfacial DMI and very weak interfacial DMI (see ref. [8]), respectively, leading to a considerable net interfacial DMI, which is beneficial for stabilizing Néel-type skyrmions. All experiments were performed at room temperature. Figure 1b shows the high angle annular dark-field scanning transmission electron microscopy (HAADF-STEM) image of the sample, revealing flat and distinct interfaces. Energy dispersive X-ray (EDX) mapping further confirms the periodical Pt/Co/Ta tri-layer structure. Figure 1c shows the out-of-plane magnetic hysteresis loop (in-plane one in Supplementary Fig. 2).

**Microscopic imaging of skyrmions**. Due to its high-spatial resolution and the ease of operation, MFM has been widely used for studying skyrmions[9,19,32]. Figure 1d–f shows the magnetic-domain structures with an out-of-plane bias magnetic fields of $B = 0$, 48, and 100 mT, respectively, where the blue (red) contrast indicates magnetization along the positive (negative) $z$ direction. At $B = 0$ mT, the sample shows a labyrinthine domain pattern (Fig. 1d), consistent with the vanishing remnant magnetization of the multilayer (Fig. 1c). Upon applying magnetic fields, the labyrinth domain transforms into skyrmions (e.g., Fig. 1e), which shrink with increasing magnetic field. At the saturation magnetic field (100 mT), the skyrmions transform into an FM single domain state (Fig. 1f). A full set of MFM images are shown in Supplementary Fig. 3. A Gaussian fitting of a typical skyrmion profile (Fig. 1g), extracted from the inset of Fig. 1g along the yellow dot line, gives a skyrmion size/diameter of 214 nm comparable to the previous reports[8,14,32,33]. Furthermore, the state of multiple isolated skyrmion was also observed (Fig. 1h) using L-TEM for sample grown on $Si_3N_4$ membrane (other images in Supplementary Fig. 4). Both the MFM and L-TEM imaging demonstrate the existence of skyrmions with small differences in skyrmion density and size due to different substrates.

**Variations of strain, interfacial DMI, and magnetic anisotropy with electric field**. Variations of strain, interfacial DMI, and magnetic anisotropy with electric field are then investigated. To obtain the strains in PMN-PT induced by electric fields, we performed in situ X-ray diffraction (XRD) measurements under different electric fields to get lattice constants $c(E)$, calculated using (002) peak by Bragg's law (raw data in Supplementary Fig. 5). Electric-field-induced out-of-plane strain (Fig. 2a), defined as $[c(E) - c(+0 \, kV/cm)]/c(+0 \, kV/cm)$, can be used to calculate the in-plane compressive strain (Supplementary Fig. 6) based on the crystal symmetry of (001)-poled PMN-PT (see ref. [34]). The out-of-plane strain is largely volatile when decreasing electric field and asymmetric for positive and negative electric fields. Moreover, the strain is close to 0.4% at $-4 \, kV/cm$, much larger than the typical value of 0.1% (see ref. [34]). More discussion is shown in Supplementary Fig. 5. The strain and magnetization under different electric fields were measured for thousands of times in previous studies[35,36], which demonstrated the stability and durability of our devices.

Interfacial DMI is essential for stabilizing skyrmions in Pt/Co/Ta (see ref. [8]), and we used Brillouin light spectroscopy (BLS)[21,22] to measure it under different electric fields. By measuring the wave-vector dependence of the frequency difference $\Delta f$ between the anti-Stokes and Stokes peaks, the interfacial DMI strength of the Pt/Co/Ta multilayer (D) can be determined by the following equation:

$$\Delta f = |f_{Anti-Stokes}| - |f_{Stokes}| = \frac{2\gamma D}{\pi M_S} k \tag{1}$$

where $k$ is the in-plane wave vector, $\gamma$ is the gyromagnetic ratio, and $M_S$ is the saturation magnetization. Figure 2b shows the wave-vector dependence of $\Delta f$ under different electric fields (raw spectra in Supplementary Fig. 7) for the $+4 \, kV/cm$ to $-4 \, kV/cm$ loop and the inset is variation of $D$ with electric field. A $D$ value of 0.772 $mJ/m^2$ can be found when removing electric field, comparable to the report[14]. The DMI value is a little bit small. However, for magnetic multilayer structures, the dipolar interaction can become significant. It has been shown that both the dipolar interaction and the DMI are important for skyrmions in magnetic multilayer structures[37]. The value of $D$ reduces from 0.772 to 0.585 $mJ/m^2$ (24% reduction) when electric field changes from $+0 \, kV/cm$ to $-4 \, kV/cm$, which is much larger than the 6% variation from 0.727 $mJ/m^2$ (at $+4 \, kV/cm$) to 0.772 $mJ/m^2$ (at $+0 \, kV/cm$). To exclude the influence of electric-field polarity on the interfacial DMI measurements, we also measured the interfacial DMI at $-1.4 \, kV/cm$ ($+4 \, kV/cm$ to $-4 \, kV/cm$ branch) because the associated strain is comparable to that of $+4 \, kV/cm$ (see $+4 \, kV/cm$ to $-4 \, kV/cm$ branch in Fig. 2a). It can be seen from the inset of Fig. 2b that the interfacial DMI 0.713 $mJ/m^2$ for $-1.4 \, kV/cm$ is comparable to the interfacial DMI value (0.727 $mJ/m^2$) of $+4 \, kV/cm$, excluding the possible influence of electric-field polarity on the interfacial DMI measurements. The similarities in the volatility and asymmetry between interfacial DMI and strain behaviors indicate a strain-mediated electric-field control of interfacial DMI. Some reports showed importance of charge accumulation/depletion[22] or ion migration[21] in controlling Rashba-type interfacial DMI (see ref. [38]) with a sizable modulation via electric-field gating on the dielectric/FM interface[21,22]. This is different from the strain-mediated modulation of the Fert–Levy DMI at the HM/FM (Pt/Co) interface[2,39] in our work, where electric-field-gating effect is negligible for HM/FM (Pt/Co) interface in our PMN-PT/Ta(4.7 nm)/[Pt/Co/Ta]$_5$ heterostructure because the charge-screening length in metallic layers is very short and the 4.7-nm-thick metallic Ta layer should effectively screen all the polarization charges at the PMN-PT surface. Thus, strain transferred from the PMN-PT to the Pt/Co/Ta multilayer is the only likely means to modulate the Fert–Levy DMI of the HM/FM (Pt/Co) interface. Specifically, the tensile out-of-plane strain transferred from the PMN-PT increases the distance between Co and Pt atom across the interface, thereby reducing the strength of the orbital hybridization among the Co–Pt–Co atoms. Since the strength of Fert–Levy DMI is proportional to the strength of such interfacial hybridization[11,12,40], the strength of the interfacial DMI of the Pt/Co interface (D) is also reduced. It should be mentioned that possible nonreciprocal effects due to the dipolar interaction were excluded in our BLS measurement (more details can be found in Supplementary Fig. 8). Overall, our work provides strong experimental evidence for a strain-mediated electric-field control of the interfacial DMI of the HM/FM interface. Such modulation of interfacial DMI strength will in turn modulate the stability of the skyrmion state.

Besides the interfacial DMI, the effective PMA of the multilayer is also critical to the control of skyrmion[41,42]. To determine the variation of the effective PMA with electric fields, we performed angular dependent FM resonance (FMR) measurements under different electric fields (see detailed experimental configuration in Supplementary Fig. 9). For each angle $\theta$, a resonance field $H_r(\theta)$ can be determined from the FMR spectrum, as shown in Fig. 2c. The effective PMA (denoted as $K_{eff}$) can then be determined by fitting $H_r(\theta)$ with the Kittel formula (see text accompanying Supplementary Fig. 9), and the fitting results are plotted by the lines in Fig. 2c. The variation of the $K_{eff}$ with electric field is shown in Fig. 2d, which resembles the variation of the interfacial DMI strength (inset of Fig. 2b) and is consistent with the strain

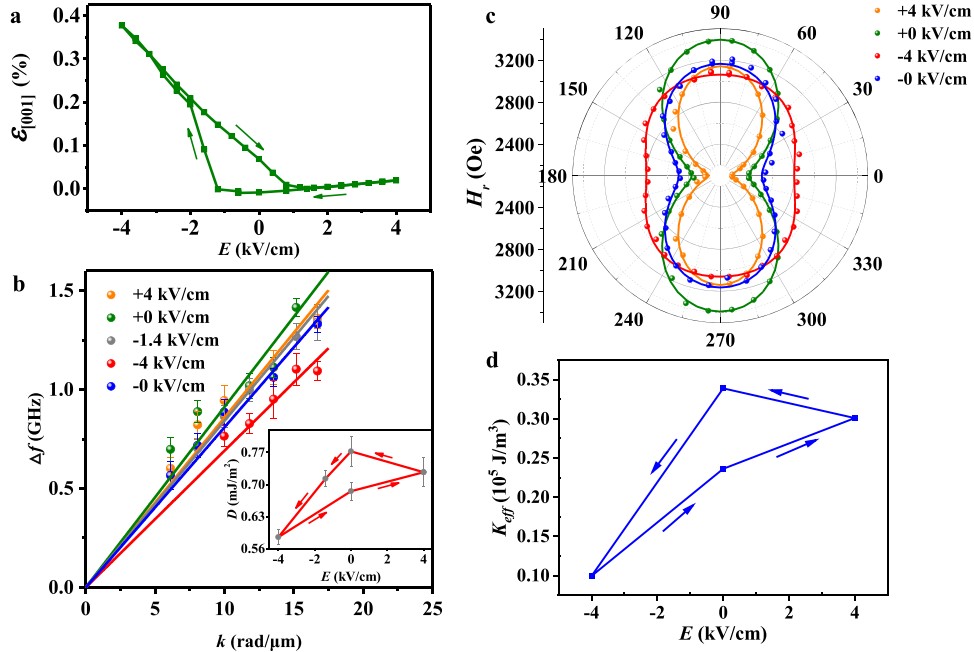

**Fig. 2 Variations of strain, interfacial DMI, and magnetic anisotropy. a** Out-of-plane strain variations of PMN-PT(001) substrate. **b** Wave-vector dependence of $\Delta f$ under different electric fields with the interfacial DMI values in the inset with the error bar obtained from the standard error of Lorentzian fitting. **c** Angle-dependent FMR resonance field $H_r(\theta)$ and corresponding Kittel formula fitting (solid lines). **d** $K_{\mathrm{eff}}$ versus electric-field curve.

variation (Fig. 2a). Thus, it can be concluded that strain transferred from the PMN-PT alters both the interfacial DMI strength of the Pt/Co interface and the effective PMA of the Pt/Co/Ta multilayer.

**Nonvolatile electric-field-induced creation of multiple skyrmions.** For investigating electric-field control of skyrmions using MFM, pretreatment procedures have been done for the sample. First, the sample was poled by varying electric fields between +4 kV/cm and −4 kV/cm for two cycles with a step of 0.2 kV/cm. Electric fields of +4 kV/cm or −4 kV/cm were then removed, yielding two remanent polarization states, denoted as +0 kV/cm and −0 kV/cm, respectively. After then, the sample was magnetically poled to saturation, followed by reducing the bias magnetic field $B_{\mathrm{bias}}$ to a lower value for subsequent in situ MFM measurement.

Figure 3 shows the electric-field-induced creation of skyrmions with $B_{\mathrm{bias}} = 60$ mT. Initially, at $E = +0$ kV/cm, the sample is almost an FM single domain (Fig. 3a), and skyrmions appear (Fig. 3b) when applying $E = −4$ kV/cm. It has been suggested that the stray magnetic field from the scanning magnetic tip can alone lead to the creation of skyrmions[32], but this possibility has been ruled out based on our control experiment (Supplementary Fig. 10). Notably, the skyrmion state remains after removing −4 kV/cm (Fig. 3c, at which $E = −0$ kV/cm). When further applying $E = +4$ kV/cm, the skyrmion state barely changes (Fig. 3d). Similar behaviors of skyrmion creation were also observed in sample starting from $E = −0$ kV/cm (Supplementary Fig. 11), that is, the creation of skyrmions can only occur when applying $E = −4$ kV/cm, under which the electric-field-induced strain from the PMN-PT is much larger than that the case of +4 kV/cm (see Fig. 2a). It was shown that the MFM images in Fig. 3a–d were taken from the same location on the sample (Supplementary Fig. 12). Moreover, the creation of skyrmions under other $B_{\mathrm{bias}}$ are shown in Supplementary Fig. 13.

We performed micromagnetic simulations to elucidate the strain-mediated electric-field creation of skyrmions in polycrystalline

[Pt/Co/Ta] × 5 multilayers, based on the strain-mediated electric-field modulation of effective PMA ($K_{\mathrm{eff}}$) and interfacial DMI strength ($D$). Since $D$ decreases sizably when sample was switched from +0 kV/cm (or −0 kV/cm) to −4 kV/cm, which by itself would impede the stabilization of skyrmions, the observed skyrmion creation/stabilization must be attributed to the modulation of $K_{\mathrm{eff}}$. Here, $K_{\mathrm{eff}} \approx K_U - 0.5\mu_0 \mathbf{M}_S^2 + \mathbf{M}_S \mathbf{B}_{\mathrm{bias}} + B_1\left(1 + \frac{2c_{12}^m}{c_{11}^m}\right)\varepsilon_{\mathrm{in-plane}}$ (see ref. [29]), where $K_U$ is the uniaxial magnetic anisotropy, $\mu_0$ is vacuum permeability, $B_1 = -1.5\lambda_S(c_{11}^m - c_{12}^m)$ is the magnetoelastic coupling coefficient of the magnetic multilayer, $c_{11}^m$ and $c_{12}^m$ are its elastic stiffness coefficient, and $\varepsilon_{\mathrm{in-plane}}$ is the isotropic in-plane strain the magnetic multilayer experiences when the underlying PMN-PT substrate deforms under applied electric fields. We use MuMax3 (version: 3.10β, which incorporates the module of magnetoelastic coupling) for simulating the evolution of local magnetization in polycrystalline Pt/Co/Ta multilayers under such isotropic in-plane strains transmitted from PMN-PT (see "Methods").

Figure 3e–h shows the simulated magnetization states at equilibrium under different in-plane biaxial compressive strains $\varepsilon_{\mathrm{in-plane}} = \varepsilon_{[110]} = \varepsilon_{[-110]}$ obtained from Supplementary Fig. 6, with $E = +0, -4, -0$, and +4 kV/cm, respectively. The average interfacial DMI strength under these electric fields, $D = 0.772$, 0.585, 0.685, and 0.727 mJ/m², respectively, are directly extracted from Fig. 2b. Overall, the simulations (Fig. 3e–h) agree well with results from MFM imaging (Fig. 3a–d). Figure 3e shows the equilibrium magnetization state under zero strain and $D = 0.772$ mJ/m² (+0 kV/cm), relaxed from a uniform upward (+z) magnetization. When applying $\varepsilon_{\mathrm{in-plane}} = -0.189\%$ and $D = 0.585$ mJ/m² (−4 kV/cm), multiple skyrmions appear (Fig. 3f). As mentioned above, since the reduction of $D$ does not favor the formation of skyrmions, the observed skyrmion creation is attributed to the reduction of the $K_{\mathrm{eff}}$ by the compressive in-plane compressive strains (note that the Pt/Co/Ta multilayer has a negative magnetostriction[24]). Being consistent with experiments, the created skyrmions can be retained (Fig. 3g) after turning off the electric field (−0 kV/cm), at which $\varepsilon_{\mathrm{in-plane}} = -0.034\%$ and

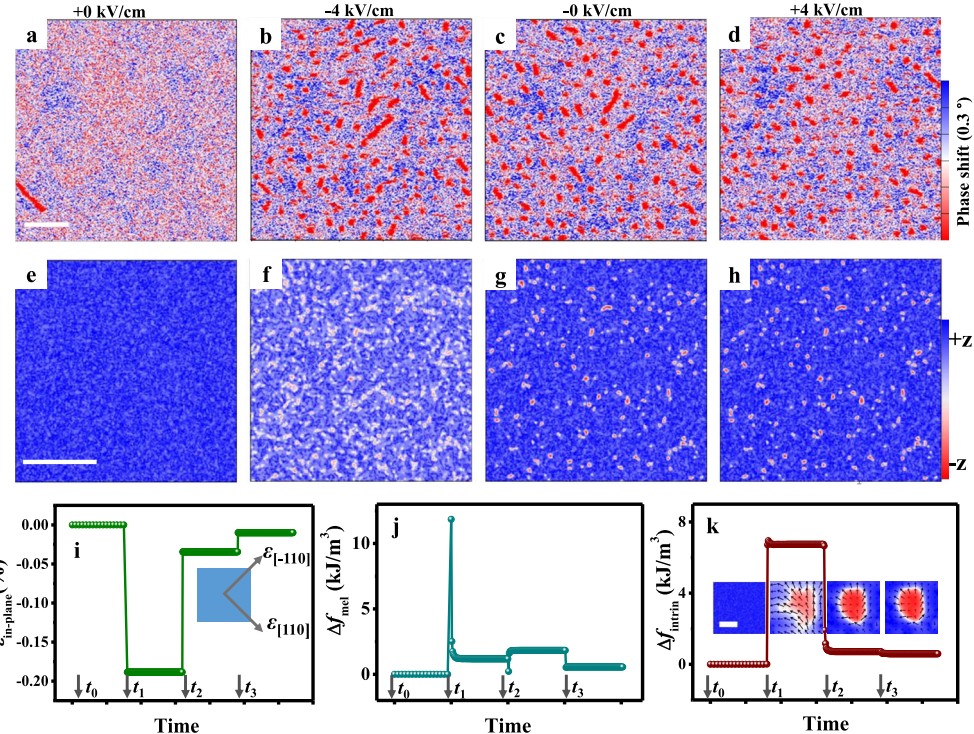

**Fig. 3 Skyrmion creation.** MFM images at $E = +0$ kV/cm (**a**), $-4$ kV/cm (**b**), $-0$ kV/cm (**c**), and $+4$ kV/cm (**d**) with $B_{bias} = 60$ mT. Corresponding simulation results of strain-mediated skyrmion creation with $\varepsilon_{[110]} = \varepsilon_{[-110]} = 0$ (initial state) and $D = 0.772$ mJ/m² (**e**), $\varepsilon_{[110]} = \varepsilon_{[-110]} = -0.189\%$ and $D = 0.585$ mJ/m² (**f**), $\varepsilon_{[110]} = \varepsilon_{[-110]} = -0.034\%$ and $D = 0.685$ mJ/m² (**g**), and $\varepsilon_{[110]} = \varepsilon_{[-110]} = -0.010\%$ and $D = 0.727$ mJ/m² (**h**), with the blue and red contrasts corresponding to magnetizations pointing up and down, respectively. The scale bar is 1 µm. Evolutions of in-plane biaxial compressive strain $\varepsilon_{in-plane} = \varepsilon_{[110]} = \varepsilon_{[-110]}$ (**i**), magnetoelastic energy density $f_{mel}$ (**j**), and intrinsic energy density $f_{intrin}$ (**k**). The time stages are $t_0$ ($E = +0$ kV/cm), $t_1$ ($E = -4$ kV/cm), $t_2$ ($E = -0$ kV/cm), and $t_3$ ($E = +4$ kV/cm), respectively. The insets of **k** are the corresponding individual skyrmion evolutions with a scale bar of 50 nm.

$D = 0.685$ mJ/m². However, neither this small remanent strain nor the slightly enhanced $D$ are the key factors for retaining the skyrmions, because the simulations shows that created skyrmions can be retained even when $\varepsilon_{in-plane}$ is set to 0 and $D$ remains to be 0.585 mJ/m² (Supplementary Fig. 14). This control study indicates that the spatially heterogenous magnetic parameters in a polycrystalline system are the key to the retainment of the created skyrmions. For example, the spatial variation of uniaxial local magnetic anisotropy mimics the presence of local pinning sites. Simulations results studying the influence of such parametric heterogeneity, average grain size, size of computation cell, and room-temperature (298 K) thermal fluctuation on skyrmion creation are provided in Supplementary Figs. 15—19. Finally, when applying $\varepsilon_{in-plane} = -0.010\%$ and $D = 0.727$ mJ/m² ($+4$ kV/cm), the overall morphology of the skyrmion state (Fig. 3h) remains largely the same compared to that in Fig. 3g, partly because the reduced $\varepsilon_{in-plane}$ and the slightly enhanced $D$ act against and in favor of stabilizing the skyrmion state, respectively.

For demonstrating that strain-mediated control of effective PMA plays a dominant role in such skyrmion creation process, Fig. 3i–k shows the evolution of $\varepsilon_{in-plane}$, magnetoelastic energy density $f_{mel}$ and intrinsic energy density $f_{intrin}$, respectively. The evolution of the constituent energy terms ($f_{anis}$, $f_{exch}$, $f_{stray}$, $f_{Zeeman}$, $f_{DMI}$, see their expressions in "Methods") of $f_{intrin}$ are shown in Supplementary Fig. 20. At $t_0$ ($E = +0$ kV/cm), the system has an almost uniform perpendicular magnetization (Fig. 3e). At $t_1$ ($E = -4$ kV/m), $f_{mel}$ increases to 11.85 kJ/m³ associated with the sudden rise of strain to $-0.189\%$. To minimize the $f_{mel}$,

perpendicular magnetization would then start to rotate toward the film plane (the magnetic easy plane). Indeed, $f_{mel}$ drops to 1.19 kJ/m³ at equilibrium. During this process, the interfacial DMI energy $f_{DMI}$ also needs to be minimized (Supplementary Fig. 20), which favors the formation of a Néel domain wall (inset in Fig. 3k). It is the simultaneous reduction of $f_{mel}$ and $f_{DMI}$ that provides the thermodynamic driving force for the creation of skyrmions (Fig. 3f). Furthermore, the energy cost is $\Delta f_{intrin} = 6.74$ kJ/m³, while energy reduction is $\Delta f_{mel} = -10.66$ kJ/m³, yielding $\Delta f_{tot} = \Delta f_{intrin} + \Delta f_{mel} = -3.92$ kJ/m³ < 0. Therefore, the skyrmion creation is favored.

Upon removing the applied large strain at $t_2$ ($E = -0$ kV/cm), $f_{mel}$ increases by 0.63 kJ/m³ because of the increase in the magnitude of perpendicular magnetization (Fig. 3g) and $f_{intrin}$ decreases by 6.03 kJ/m³. Moreover, $f_{intrin}$ increases by 0.71 kJ/m³ compared with the initial state ($t_0$), suggesting that the system in which skyrmions are maintained (Fig. 3g) is metastable. Similarly, further reducing the compressive strain from $-0.034$ to $-0.01\%$ and increasing $D$ slightly from 0.685 to 0.727 mJ/m² at $t_3$ ($E = +4$ kV/cm), the corresponding $f_{mel}$ and $f_{DMI}$ both decreases, but the former shows a more sizable change (see Supplementary Fig. 20), indicating a stronger strain effect. Since the reduction of an in-plane compressive strain (hence larger $K_{eff}$) reduces the stability of skyrmions, some smaller skyrmions annihilate. From the magnetization vector distributions in the local region of the multilayer (insets in the bottom panel), it can be seen that most of the magnetization vectors near the domain wall are perpendicular to the wall, directing from inside of the skyrmion to the outside. This indicates a DMI-stabilized left-chirality Néel wall, which is

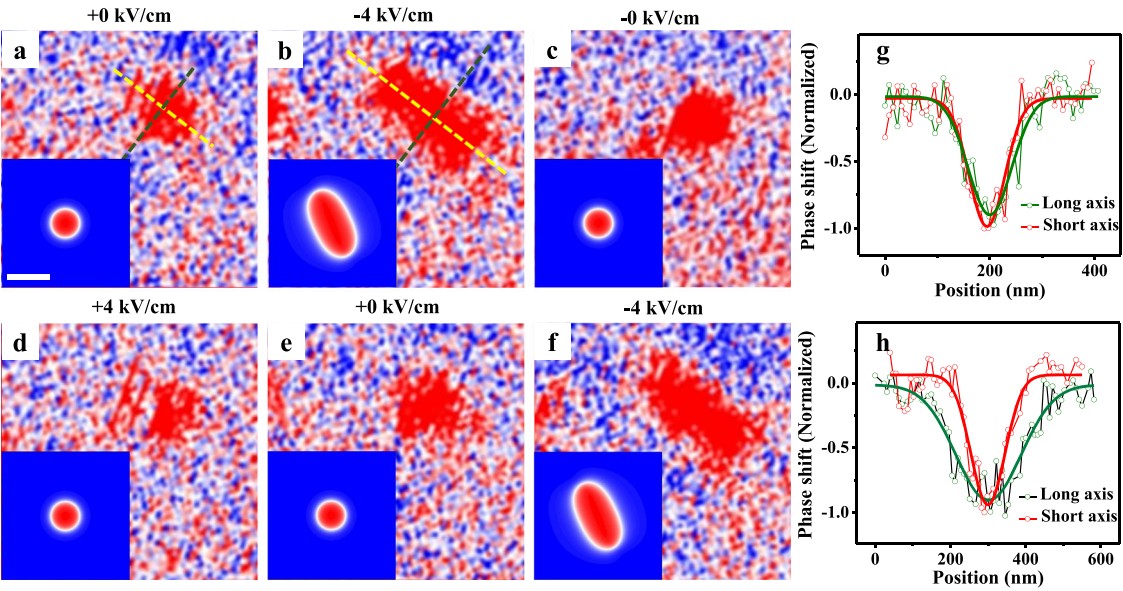

**Fig. 4 Skyrmion deformation.** Isolated skyrmion morphology at $E = +0$ kV/cm (**a**), $-4$ kV/cm (**b**), $-0$ kV/cm (**c**), $+4$ kV/cm (**d**), $+0$ kV/cm (**e**), and $-4$ kV/cm (**f**) with $B_{bias} = 55$ mT. **g**, **h** Skyrmion line-cut profiles along the major and minor axes in **a** and **b** with Gaussian fittings (solid lines), respectively. The insets in **a**–**f** show the simulation results of strain-mediated deformation of one single skyrmion with $\varepsilon_{[-110]} = \varepsilon_{[110]} = 0$ $D = 0.772$ mJ/m$^2$ for $+0$ kV/cm (**a**), $\varepsilon_{[-110]} = -0.169\%$, $\varepsilon_{[110]} = 0$, $D = 0.585$ mJ/m$^2$ for $-4$ kV/cm (**b**), $\varepsilon_{[-110]} = \varepsilon_{[110]} = -0.023\%$ $D = 0.685$ mJ/m$^2$ for $-0$ kV/cm (**c**), and $\varepsilon_{[-110]} = \varepsilon_{[110]} = -0.012\%$ $D = 0.727$ mJ/m$^2$ for $+4$ kV/cm (**d**). The scale bar is 100 nm.

further confirmed by our asymmetric domain expansion study using L-TEM (Supplementary Fig. 21).

**Deformation and annihilation of isolated skyrmions**. We also investigated deformation and annihilation of isolated skyrmions using MFM, with a high-spatial resolution via a smaller scanning size (2 μm), which is likely unperceived in Fig. 3 due to the larger scanning size (5 μm). The sample was in the remanent polarization state after positive electric poling ($E = +0$ kV/cm) with $B_{bias} = 55$ mT. Figure 4 demonstrates the anisotropic skyrmion deformation induced by in situ electric fields. The anisotropic deformation factor $f$ is defined as $f = 1 - r_1/r_2$, with $r_1$ and $r_2$ the minor and major axes of the elliptic skyrmion, respectively. In the initial state ($E = +0$ kV/cm), the skyrmion is circular (Fig. 4a) with the line-cut profiles shown in Fig. 4g. At $-4$ kV/cm, the skyrmion exhibits evident anisotropic deformation (Fig. 4b) along the diagonal direction of magnetic multilayer, i.e., the [110] direction of PMN-PT, with $f \sim 48\%$ ($r_1 = 180 \pm 12$ nm, $r_2 = 346 \pm 22$ nm) deduced from Fig. 4h. At $-0$ kV/cm (after removing $-4$ kV/cm), the skyrmion becomes circular again with a size similar to its initial state. A subsequent application of $+4$ kV/cm (Fig. 4d) followed by its removal ($+0$ kV/cm, Fig. 4e), there is no appreciable deformation. However, the deformation occurs again upon applying $-4$ kV/cm (Fig. 4f), suggesting that the deformation is repeatable and appears only under $-4$ kV/cm. This is consistent with the earlier-discussed observation that the creation of multiple skyrmions also occurs only under $-4$ kV/cm, where both the strain and interfacial DMI show the most significant change (Fig. 2a, b). Note that the phase contrast in Fig. 4d is somewhat pale, which may be attributed to the influence of magnetic tip[32], but this does not contradict the main finding.

The experimental work on skyrmion deformation is rather limited[43]. A deformation factor of 30% was reported for B20 compound FeGe under a uniaxial strain of 0.3% (see ref. [43]). This exotic emergent elasticity of skyrmions in FeGe was attributed to either the anisotropic magnetoelastic energy[44,45] or the anisotropic DMI strength[43] induced by the uniaxial strain. In magnetic

multilayers with interfacial DMI, there is only one computational report on such emergent elasticity of isolated skyrmion[46], which was attributed to anisotropic magnetoelastic energy.

In our sample, although the average in-plane strains from a [001]-poled (001)PMN-PT crystal are expected to be isotropic based on its crystal symmetry[34], the local in-plane strain at the PMN-PT surface, arising from the 109° ferroelectric domain switching[25,36], should be anisotropic. It is such local strain anisotropy that leads to the deformation of a single local skyrmion, which is much smaller than the micron-sized ferroelectric domain at the PMN-PT surface. Specifically, such local anisotropic strains, which are along the [110] and [−110] axes of the PMN-PT, would lead to anisotropic Néel wall energy $\sigma_w = 4\sqrt{A_{ex}K_{eff}} - \pi|D|$ (refs. [8,20]) along these two axes by modulating either the local magnetoelastic anisotropy (which contributes to $K_{eff}$) or the local DMI strength $D$, or both. Here, $A_{ex}$ is the exchange stiffness. Regardless of the details, the skyrmion radius is always smaller along the axis with a lower $\sigma_w$ (see ref. [46]). This is analogous to Wulff construction: lower surface energy (wall energy) of a crystal (skyrmion) yields shorted vector length (skyrmion radius) at thermodynamic equilibrium.

With these in mind, we simulate the deformation of one single skyrmion by applying biaxial in-plane anisotropic strains. The anisotropic magnetoelastic energy $f_{mel}$ was automatically considered (see expression of $f_{mel}$ in "Methods"), but the local DMI strength $D$ was assumed to be isotropic for simplicity. The simulation results, which agree well with experimental observations, are shown in the insets of Fig. 4.

We also observed repeatable annihilation and creation of isolated skyrmion (Fig. 5). The initial state (Fig. 5a) shows the skyrmion with irregular shape implying presence of local inhomogeneity. At $E = -4$ kV/cm, there is no phase contrast (Fig. 5b), indicating the annihilation of skyrmion. Upon the removal of this negative poling field, the skyrmion reappears at $E = -0$ kV/cm (Fig. 5c). Then, applying $+4$ kV/cm, the skyrmion shrinks in size (Fig. 5d). After removing this positive poling field (now at $E = +0$ kV/cm), the skyrmion returns to its initial state

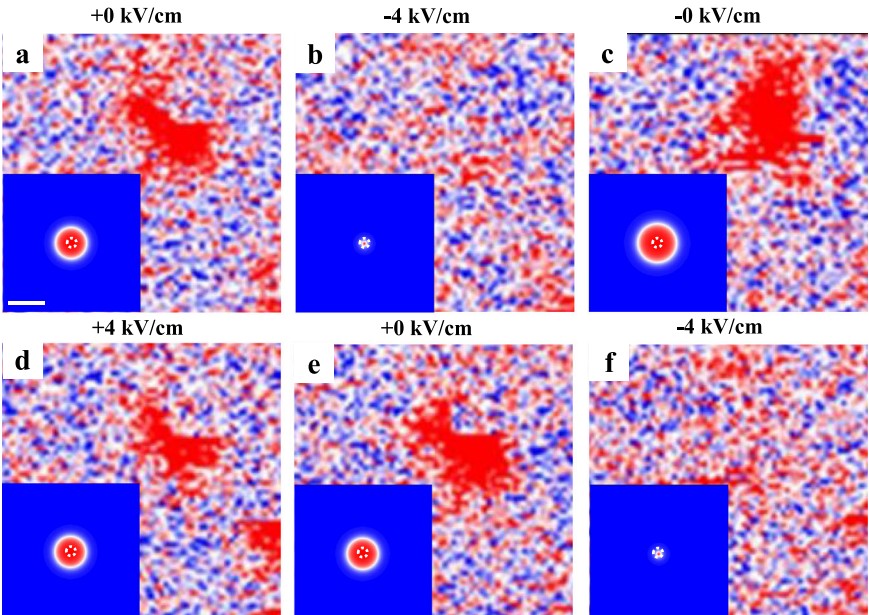

**Fig. 5 Skyrmion creation/annihilation.** Isolated skyrmion morphology at $E = +0$ kV/cm (**a**), $-4$ kV/cm (**b**), $-0$ kV/cm (**c**), $+4$ kV/cm (**d**), $+0$ kV/cm (**e**), and $-4$ kV/cm (**f**) with $B_{bias} = 55$ mT. The insets in **a**–**f** show the simulation results of strain-mediated annihilation and reappearance of one single skyrmion with $\varepsilon_{[-110]} = \varepsilon_{[110]} = 0$, $D = 0.772$ mJ/m$^2$ for $E = +0$ kV/cm (**a**, **e**); $\varepsilon_{[-110]} = \varepsilon_{[110]} = -0.0425\%$, $D = 0.585$ mJ/m$^2$ for $E = -4$ kV/cm (**b**, **f**); $\varepsilon_{[-110]} = \varepsilon_{[110]} = -0.0415\%$, $D = 0.685$ mJ/m$^2$ for $E = -0$ kV/cm (**c**); and $\varepsilon_{[-110]} = \varepsilon_{[110]} = -0.012\%$, $D = 0.727$ mJ/m$^2$ for $E = +4$ kV/cm (**d**). A 20-nm diameter pinning site with ~5% lower perpendicular anisotropy was specified, as indicated by the dashed circle in the center. The scale bar is 100 nm.

(Fig. 5e). Applying $-4$ kV/cm again, the skyrmion annihilates again (Fig. 5f).

The insets in Fig. 5 show the simulation results for the processes discussed above. Notably, for simulating the reappearance of the skyrmion from exactly the same location where it annihilates, we found that it is necessary to introduce a small pinning site (20 nm in diameter herein) with moderately (~5%) lower perpendicular anisotropy. The critical role of lower-anisotropy pinning site in achieving localized skyrmion deletion/re-creation has been demonstrated in systems with Rashba-type DMI arising from the FM/dielectric interface[19]. As shown in the inset of Fig. 5b, this lower-anisotropy pinning site leads to the stabilization of a tiny (diameter ~10 nm) skyrmion, which is too small to discern with our MFM (with a spatial resolution of ~10 nm as well), yet can function as the nucleus for the reappearance (growth) of skyrmion at exactly the same location after the removal of electric field (the inset of Fig. 5c). Our simulation results suggest the possibility of harnessing pinning sites for realizing a spatially precise control of skyrmions in continuous magnetic layer, which could be useful for device applications.

It is worth noting that there is an existing report on tuning the density and stability of the skyrmions in BaTiO$_3$/SrRuO$_3$ bilayer heterostructures by ferroelectric polarization switching[47]. In that work, voltage-controlled topological Hall effect (THE) measurement to provide an indirect evidence for skyrmion manipulation at low temperatures (<100 K), but a direct observation via in situ imaging is missing. Moreover, the origin of the so-called THE signal in SrRuO$_3$ is still an open question[48,49]. After completing our manuscript, we noticed a very recent report on creation of micron-sized skyrmion bubbles (3–6 µm) with in-plane radiofrequency voltages by employing surface acoustic waves in multilayers of Pt/Co/Ir grown on LiNbO$_3$, where it was found that static electric fields hardly affect skyrmions[50]. By contrast, in our work, we show manipulations of nanometer-scale skyrmions by using static and vertical electric fields. The use of static and vertical electric field is generally more compatible with existing manufacturing platform of spintronics, and is more energy efficient.

## Discussion

In conclusion, we have demonstrated the electric-field control of skyrmions, especially the first observation of nanoscale skyrmion creation and Fert–Levy DMI variations, in FM/ferroelectric multiferroic heterostructures. Electric-field-induced variations of both the magnetic anisotropy (by FMR) and the interfacial DMI at the Pt/Co interface (by BLS) are consistent with the trend of electric-field-induced strain (by XRD), indicating that the electric-field control of both quantities is strain mediated. Specifically, the experimentally observed electric-field control of the Pt/Co interface DMI is explained by the weakening of Pt–Co hybridization at the interface induced by the out-of-plane tensile strain. Furthermore, we demonstrate nonvolatile electric-field creation of skyrmions. Micromagnetic simulations indicate the dominant role of strain-mediated control of magnetoelastic anisotropy (with respect to the control of interfacial DMI) in skyrmion creation, as well as the importance of polycrystal inhomogeneity (specifically, the local pinning sites) in retaining these metastable skyrmions at zero electric fields. We also observe electric-field-induced large and reversible deformation (up to 48%), repeatable annihilation, and reappearance of one isolated skyrmion. These observations are also interpreted using micromagnetic simulations. These findings could stimulate more research into the strain-mediated electric-field control of skyrmions or other topologically nontrivial spin textures, and pave the way toward more energy-efficient skyrmion-based spintronic devices.

## Methods

**Sample preparation**. PMN-PT FE single-crystal substrates are (001)-cut and one-side polished with a size of $5 \times 3 \times 0.5$ mm$^3$. Ta(4.7 nm)/[Pt(4 nm)/Co(1.6 nm)/Ta (1.9 nm)] $\times$ 5 multilayers were grown by magnetic sputtering at room temperature with a base pressure of $1 \times 10^{-6}$ Pa without a magnetic field. The first layer, Ta, was used to flat the surface of PMN-PT and promote the interface quality of Pt/Co. The top Ta layer is a capping layer preventing oxidation of Co and also as the top electrode. Au layer with a thickness of 300 nm was deposited on the bottom of PMN-PT as the electrode.

**Magnetic property measurement system**. Out-of-plane and in-plane magnetic hysteresis loops were measured with a magnetic property measurement system (MPMS 7 T; Quantum Design).

**Magnetic force microscopy**. MFM was performed in ambient conditions with an Infinity Asylum Research AFM. The stray field of magnetic tip will inevitably interact with magnetic domain and distort them[32]. To reduce the influence of magnetic tip on domain, an optimized magnetic tip (SSS-MFMR)[51] has been used in our work, which has a radii of 30 nm with a low coercivity ~ 12 mT and ultralow magnetic moment ~ $8 \times 10^4$ A/m. MFM images were taken in tapping/lift mode with a lift height of 30 nm.

The safety voltage for MFM is 200 V. Therefore, we apply ±4 kV/cm to the sample with a thickness of 0.5 mm.

**Lorentz transmission electron microscopy**. L-TEM imaging was carried out on Lorentz TEM (JEOL 2100F) under perpendicular magnetic fields by gradually increasing the objective lens current.

**Scanning transmission electron microscopy**. STEM and EDS mapping were performed on an ARM-200F (JEOL) operated at an acceleration voltage of 200 kV and equipped with double-spherical aberration (Cs) correctors. The TEM sample was prepared by using a focused ion beam with the cross-sectional lamella thinned down to 100 nm.

**X-ray diffraction**. XRD was operated on a Rigaku D/max-RB X-ray diffractometer with Cu Kα radiation.

**Brillouin light spectroscopy**. The principle of BLS is that[52] the wave-vector direction of the magnetostatic surface spin wave in FM layer has a spatial chirality depending on the cross products of the normal direction and the direction of the magnetization, thus a pair of symmetric Stokes and anti-Stokes peaks are present in the BLS spectra corresponding to a fixed wave vector but opposite direction. However, the interfacial DMI favors certain spatial chirality same with only one of the wave vector, which induces an asymmetric modification of the frequency dispersion relation of Stokes and anti-Stokes peaks. The wave-vector dependence of the frequency difference between the Stokes and anti-Stokes peaks, $\Delta f$, has a linear relation with the in-plane wave vector $k$, and the slope is determined by $D$, see Eq. (1).

BLS was performed by using a single-mode solid-state laser with a wavelength of 532 nm and a power of 30 mW. The 180°-backscattered light was analyzed using a JRS Sandercock-type multipass tandem Fabry–Pérot interferometer. An external magnetic field of 500 mT was applied parallel to the sample surface plane and perpendicular to the in-plane wave vector (DE mode configuration).

**FMR measurement**. FMR measurements were performed by using the JEA-FS200 ESR spectrometer of JEOL Co., equipped with an X-band microwave generator with a frequency of 9.1 GHz and a cylindrical microwave resonant cavity. The sample was mounted to a rotating stage with the magnetic field direction unchanged.

**Micromagnetic simulation**. We use MuMax[3] (see ref. [53]) (version: 3.10β, which incorporates the module of magnetoelastic coupling) for simulating the evolution of local magnetization in polycrystalline [Pt(4 nm)/Co(1.6 nm)/Ta(1.9 nm)] × 5 multilayers under voltage-induced strains transmitted from the underlying piezo-electric PMN-PT substrate. In the simulations, the total free energy density $f_{tot} = f_{anis} + f_{exch} + f_{stray} + f_{Zeeman} + f_{mel} + f_{DMI}$ is contributed by the following energy densities: the magnetocrystalline uniaxial anisotropy energy density $f_{anis} = -K_u(\mathbf{u} \cdot \mathbf{m})^2$, where $\mathbf{u}$ is the unit vector of the uniaxial anisotropy and $\mathbf{m}$ is the normalized local magnetization, $K_u$ is the uniaxial anisotropy constant; the magnetic exchange energy density $f_{exchange} = A_{ex}(\nabla \mathbf{m})^2$, where $A_{ex}$ is magnetic exchange coupling coefficient; the magnetostatic stray field energy density $f_{stray} = -\frac{1}{2}M_S\mathbf{m} \cdot \mathbf{B}_{stray}$, where $M_S$ is the saturation magnetization and $\mathbf{B}_{stray}$ is the magnetostatic stray field; the Zeeman energy density associated with the externally applied magnetic field $\mathbf{B}_{ext}$, i.e., $f_{Zeeman} = -M_S\mathbf{m} \cdot \mathbf{B}_{ext}$; the magnetoelastic energy density $f_{mel} = B_1(\varepsilon_{xx}m_x^2 + \varepsilon_{yy}m_y^2 + \varepsilon_{zz}m_z^2) + 2B_2(m_xm_y\varepsilon_{xy} + m_xm_z\varepsilon_{xz} + m_ym_z\varepsilon_{yz})$, where $B_1$ and $B_2$ are magnetoelastic coupling coefficients, but the $B_2$-related term can be omitted in our case because average shear strains are set to be zero ($\varepsilon_{xy} = \varepsilon_{xz} = \varepsilon_{yz} = 0$); the interfacial DMI energy density $f_{DMI} = D[m_z(\nabla \cdot \mathbf{m}) - (\mathbf{m} \cdot \nabla)m_z]$, where $D$ is the interfacial DMI strength. The variation of $f_{tot}$ with respect to the change in $\mathbf{m}$ gives rise to the effective magnetic field $\mathbf{B}_{eff} = -\frac{1}{M_S}\frac{\delta F_{tot}}{\delta \mathbf{m}}$, which drives the motion of the local magnetization $\mathbf{m}$ in a way described by the Landau–Lifshitz–Gilbert equation. The influence of thermal fluctuation at room temperature (298 K) on the switching of $\mathbf{m}$ can be considered by adding a thermal fluctuation field $\mathbf{B}_{therm}$ to the total effective magnetic field $\mathbf{B}_{eff}$, that is, $\mathbf{B}_{eff} = -\frac{1}{M_S}\frac{\delta F_{tot}}{\delta \mathbf{m}} + \mathbf{B}_{therm}$. Here, $\mathbf{B}_{therm} = \eta\sqrt{\frac{2\alpha k_B T}{M_S\gamma\Delta V\Delta t}}$, where $\alpha$ is the damping parameter, $k_B$ is Boltzmann constant, $T$ is temperature, $\gamma$ is the gyromagnetic ratio,

$\Delta V$ is the volume of simulation cell, and $\Delta t$ is time step in real unit, $\beta = (\eta_x, \eta_y, \eta_z)$, where $\eta_x, \eta_y, \eta_z$ are random numbers that have a standard normal distribution and a mean of zero, and are uncorrelated in both space and time. It is found that adding $\mathbf{B}_{therm}$ (with $T = 298$ K) yields overall similar behaviors of strain-mediated skyrmion creation to the case where $\mathbf{B}_{therm} = 0$ (by setting $T = 0$ K), as shown in Supplementary Fig. 19. The simulations shown in the main paper are performed at $\mathbf{B}_{therm} = 0$ for investigating and analyzing the energetics of the skyrmion switching more clearly.

The simulations simultaneously consider strain-induced changes in both the magnetic anisotropy and the interfacial DMI strength $D$. Unless stated otherwise, the former was considered by using average strains measured using strain gauge under different electric fields (Fig. 2a), which are related to the magnetoelastic energy density $f_{mel}$. The latter was considered by using the $D$ values measured using BLS under different electric fields (Fig. 2b), which are relate to the interfacial DMI energy density $f_{DMI}$. The influence of strain on the magnetic anisotropy can be interpreted in a way that the strain is modulating the effective PMA of the multilayer as discussed in the main text.

The multilayer is described by 3D discretized computational cells of $n_{x'}\Delta x' \times n_{y'}\Delta y' \times 9\Delta z'$, where $n_{x'}$ and $n_{y'}$ are numbers of computational cells, and $\Delta x'$, $\Delta y'$, and $\Delta z'$ are cell sizes. Along the thickness direction ($z'$ axis), the bottom five layers and top two layers of grids with $\Delta z' = 0.8$ nm are set as nonmagnetic materials to describe the Pt and Ta layer, while the middle two layers describe the magnetic Co layer. In-plane periodic boundary conditions are applied by adding one repetition of the simulation system along both $x'$ and $y'$ axes. Five repetitions of the simulation system are set along $z'$ axis to describe five repetitions of the Pt/Co/Ta tri-layer. The coordinate systems used in the experiment and simulations are different, shown in Supplementary Fig. 1b. Specifically, we only consider the strain along the diagonal direction of PMN-PT(001), and $\varepsilon_{x'x'}^m = \varepsilon_{[110]}^{PMN-PT} = \varepsilon_{[110]}$, $\varepsilon_{y'y'}^m = \varepsilon_{[-110]}^{PMN-PT} = \varepsilon_{[-110]}$ as discussed later. We use $\varepsilon_{in-plane} = \varepsilon_{[110]} = \varepsilon_{[-110]}$ in the main text.

For simulating strain-mediated electric-field control of multiple skyrmions, we first relax an initially uniform [001] magnetization under a bias external magnetic field $B_{bias}$. The obtained equilibrium magnetization distribution is used as the initial state. We then apply a spatially uniform strain to the entire magnetic multilayer. A complete transmission of the voltage-induced strain from the PMN-PT surface to the magnetic multilayer is considered[26], yielding $\varepsilon_{x'x'}^m = \varepsilon_{[110]}^{PMN-PT}$, $\varepsilon_{y'y'}^m = \varepsilon_{[-110]}^{PMN-PT}$.

The out-of-plane strain of the magnetic layer $\varepsilon_{z'z'}^m$ is calculated by $\varepsilon_{z'z'}^m = -\frac{c_{12}^m(\varepsilon_{x'x'}^m + \varepsilon_{y'y'}^m)}{c_{11}^m}$ according to the thin-film boundary condition.

For the simulation of skyrmions creation in Fig. 3, we describe the system as a polycrystal considering the heterogeneity in a polycrystalline sample, with a mean grain size of 20 nm, $\Delta x' = \Delta y' = 3$ nm, and $n_{x'} = n_{y'} = 1000$ and $B_{bias} = 60$ mT. We define grain-like regions with region-specific uniaxial magnetic anisotropy constant ($K_U$), uniaxial magnetic anisotropy direction, and interfacial DMI strength ($D$) using Voronoi tessellation. Specifically, we generate 256 different $K_U$ obeying a uniform distribution around $K_{U0} = 6.5 \times 10^5$ J/m³ (based on our own measurement, see Supplementary Fig. 2) within the range of ($0.9–1.1K_{U0}$), 256 different $\theta$ (the angle between the axis of uniaxial magnetic anisotropy and the $z'$-axis) obeying a uniform distribution around 0° within the range of ($-20°–20°$), 256 different $\varphi$ (the angle between $+x'$-axis and the projection of axis of uniaxial magnetic anisotropy on the $x'y'$ plane) obeying uniform distribution within the range of (0°–360°), and 256 different $D$ obeying a uniform distribution around $D_0$ within the range of ($0.9–1.1D_0$). For each parameter set ($K_U$, $\theta$, $\varphi$, $D$), the 256 different numbers are randomly assigned to the grain-like regions. Note that the $D$ is the averaged interfacial DMI strength of the entire sample, which is taken from our own experimental measurement and dependent on the applied electric fields (see inset of Fig. 2b). Furthermore, for a (001)-oriented PMN-PT single crystal, the average in-plane strain induced by electric field is expected to be biaxial isotropic[54], i.e., $\varepsilon_{[110]}^{PMN-PT} = \varepsilon_{[-110]}^{PMN-PT} = \varepsilon_0^{PMN-PT}$. Assuming volume conservation, $\varepsilon_0^{PMN-PT}$ can be calculated from the experimentally measured $\varepsilon_{[001]}^{PMN-PT}$, which is the average out-of-plane strain induced by electric field, via $\varepsilon_0^{PMN-PT} = \nu_{PMN-PT} \cdot \varepsilon_{[001]}^{PMN-PT}$, with Poisson's ratio $\nu_{PMN-PT} = -0.5$ (see ref. [34]), shown in Supplementary Fig. 6.

However, given the local strain variations on the PMN-PT surface caused by spatially heterogeneous ferroelectric domain switching[25,36], we introduce anisotropic in-plane strain to simulate the deformation and annihilation of one single skyrmion as discussed in the Figs. 4 and 5 and Supplementary Fig. 22. In these cases, we use $\Delta x' = \Delta y' = 0.5$ nm and $n_{x'} = n_{y'} = 600$ as well as spatially homogenous magnetic parameters ($K_U = 6.5 \times 10^5$ J/m³; $D$ varies with electric fields) for simplicity; the bias magnetic field $B_{bias} = 30$ mT. Under this setup, a circular-shaped 68-nm diameter skyrmion can be stabilized under zero strain. Anisotropic in-plane strains are then applied.

The rest of the magnetic parameters are all set to be spatially uniform in both the polycrystal and single-crystal (i.e., spatially uniform $K_u$ and $D$) model. These parameters include: saturation magnetization $M_S = 9.48 \times 10^5$ A/m (from our own experimental measurement), magnetic exchange coupling coefficient $A_{ex} = 1.0 \times 10^{-11}$ J/m (see ref. [14]), magnetostriction constant $\lambda_S = -180$ ppm (which leads to best fit of the experimentally measured skyrmion morphology), and elastic stiffness coefficients $c_{11}^m = 290.86$ GPa and

$c_{12}^m = 160$ GPa, which are calculated from thickness-weighted average of the Young's modulus and Poisson's ratio of the Pt, Co, and Ta layers[55]. Young's modulus are $Y_{Pt} = 160$ GPa, $Y_{Co} = 210$ GPa, and $Y_{Ta} = 186$ GPa, and Poisson's ratio are $\nu_{Pt} = 0.38$, $\nu_{Co} = 0.31$, and $\nu_{Ta} = 0.34$ (https://periodictable.com/Properties/A/YoungModulus.html).

**Note added to proof**. After submission of this work and during the period of preparation for the response to the reviewers' comments, an article reporting study on electric-field control of skyrmions in multiferroic ferromagnetic/ferroelectric nanostructures[56] has appeared. We focus on skyrmions in continuous thin films, while they focus on nanopatterned confined skyrmions and the behaviors of the skyrmions in these two cases are different.

## Data availability

The data that support the findings of this study are available from the corresponding author upon reasonable request.

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

## Acknowledgements

This work was supported by the Science Center of National Science Foundation of China (Grant No. 51788104) and National Science Foundation of China (Grant No. 51831005). J.-M.H. acknowledges support from the National Science Foundation under award CBET-2006028, and a start-up fund from the University of Wisconsin-Madison. Micromagnetic simulations were performed using Bridges at the Pittsburgh Super-computing Center through allocation TG-DMR180076, which is part of the Extreme Science and Engineering Discovery Environment (XSEDE) and supported by NSF grant ACI-1548562. W.J. acknowledges support from Tsinghua University Initiative Scientific Research Program and the Beijing Advanced Innovation Center for Future Chip (ICFC). The authors acknowledge X. Yuan for support for MFM data difference process, Y. Liang for the helpful discussion of MFM measurement, and Y. Zhou for discussion of PMN-PT ferroelectric domain switching. The authors acknowledge C. Feng, W. Sun, and Q. Liu for useful discussion during this work.

## Author contributions

Y.Z. and Y.B. conceived the project and designed the research. W.J. and H.Z. planed and deposited the multilayers. Y.B. performed the MPMS, XRD, and MFM measurement with the help of Yike.Z. and Y.W. Authors M.C., J.-M.H., and C.N. gave help for the MFM measurement. G.C., Y.G., and Yike.Z. carried out the BLS measurement. Ying.Z. performed the L-TEM measurement. H.T. contributed to the STEM measurement. H.D. measured asymmetric domain expansion under in-plane magnetic field by L-TEM. S.Z. and J.-M.H. performed the micromagnetic simulation. Yike.Z., W.S., Y.W., and Q.L. performed the FMR measurement. Y.B., S.Z., Yike.Z., Y.W., J.-M.H., and Y.Z. prepared the manuscript with contributions from all coauthors.

## Competing interests

The authors declare no competing interests.

## Additional information



