## [Peer Review File · Nature Communications]

Editorial Note: Parts of this peer review file have been redacted as indicated to remove third-party material where no permission to publish could be obtained.

REVIEWER COMMENTS

Reviewer #1 (Remarks to the Author):

Review on NCOMMS-20-21660-T : Electric-field manipulation of skyrmion in multiferroic heterostructure via magnetoelectric coupling

Manuscript ID : NCOMMS-20-21660-T

Author(s): You Ba et al.

Authors demonstrated electric-field control of skyrmion through strain-mediated magnetoelectric coupling in multiferroic heterostructure. They showed the process of skyrmions creation, isolated skyrmion deformation and annihilation by performing magnetic force microscopy with in situ electric fields. In the present research work the authors confirmed the decisive role of magnetoelastic coupling in skyrmion manipulation through micromagnetic simulations. Although the fact that there are several scientific simulation works on creating, deleting and driving skyrmions the present work is the first experimental work in FM/FE multiferroic heterostructure that is being reported so far providing so a platform for the research of electric-field control of skyrmion in multiferroic heterostructure.

The paper is well organized and the presentation of the results is clear. The conclusions are soundly supported by data presented in the manuscript and in the supplementary data. The paper is very interesting, well written and provides concise explanations, justifications and physical insights towards new directions regarding electric-field manipulation of skyrmion in multiferroic heterostructure via magnetoelectric coupling.

I recommend this paper for publication in "Nature Communications " subject to a number of minor changes and corrections that could be addressed. In order to improve readability I suggest the following:

1. The authors should try to include some figures (e.g. Fig. 1: Schematic of the sample configuration, Supplementary Fig. 18: A series of skyrmion morphology under different in-plane biaxial tensile strain.) in order to increase the readability of the paper.

2. The present study and the experimental procedure took place around room temperature. The authors in their micromagnetics simulations presented in Methods Section they refer to the use of "mumax 3 (version: 3.10 β , which incorporates the module of magnetoelastic coupling) for simulating the evolution of local magnetization in polycrystalline [Pt (4 nm)/Co (1.6 nm)/Ta (1.9 nm)] \times 5 multilayers under voltage-induced strains transmitted from the underlying piezoelectric PMN-PT substrate". It would be nice for the readability and the reproducibility of the paper to include the actual energy functional used having contributions for instance from exchange, anisotropy, demagnetization and Dzyaloshinski-Moriya or from thermal fields if we refer to temperatures differing from 0K. Presumably (unfortunately is not clear in the manuscript), the micromagnetic simulations took place at 0K. The authors should comment on this in a more detailed manner since they compare results from simulations at 0K with experimental measurements at elevated temperatures.

Reviewer #2 (Remarks to the Author):

The authors show experimental indications for creation, deformation and annihilation of skyrmions in ferromagnetic multilayers by uniform elastic strain. The strain is caused by application of a voltage to piezoelectric PMN-PT substrates, such that an indirect electric field control of the above processes is achieved.

The authors explain their experimental observations by assuming that the strain changes both, magnetic anisotropy and DMI. I found the extraction of the DMI-modification by electric fields, which

is then used as input for micromagnetic simulations, unconvincing. My detailed reasoning is the following:

As shown in Fig. 3a, electric-field control of strain is hysteretic and quite unusual in their particular sample (same sign of strain for positive/negative electric fields, large asymmetry). In Fig. 3a, Fig. 2 and Fig. 4, the authors use electric fields in the range ± 4 kV/cm. However, in Fig. 3b, the authors use ± 8 kV/cm. The authors then assume that strain, electric field and DMI are all linearly related to extract DMI at ± 4 kV/cm from Fig. 3b, which is then used in their micromagnetic simulations. This leads to the following inconsistency:

Fig. 3a shows that positive/negative electric fields lead to drastically different strains. For $E > 0$, the strain is very small (factor 10 smaller than for negative E). However, in Fig. 3b, for $E = \pm 8$ kV/cm, similar DMI is found. From this figure the authors then extract similar DMI for ± 4 kV/cm (0.73 and 0.76 mJ/m²) by linear interpolation. The finding of such similar values of DMI for different E polarities is inconsistent with a strain-based origin of the DMI modification, considering that Fig. 3a demonstrates very different strain for ± 4 kV/cm. Notably, in Fig. 4d the authors observe no skyrmion deformation with $E > 0$, consistent with Fig. 3a but inconsistent with Fig. 3b.

The authors need to perform the DMI extraction experiments using the same electric field hysteresis loops (i.e. from $+4$ kV/cm to -4 kV/cm and not from $+8$ kV/cm to -8 kV/cm) as used for the rest of the study if they want to extract meaningful DMI values as input for their micromagnetic simulations. With the data present in the manuscript, the assumption that the DMI is essentially unchanged under electric field polarity reversal (Fig. 3b) is inconsistent with their further observations.

Furthermore, I also found the data in Fig. 2 somewhat inconclusive. First, the authors should clarify how they made sure that Fig. 2 a-d all show the exact same location on the sample. Second, in Fig. 2, there does not appear to be a clear correlation between the domains/skyrmions in panels b,c,d. This is different to the micromagnetic simulations, where one can identify that a large number of skyrmions are unchanged between panels g and h. If the sample stays in a skyrmion state when switching off the voltage but individual skyrmions are not retained, this does not seem to be very useful for memory applications and it also seems to contrast the findings in Fig. 5. Also, the authors state on page 9, line 190 "Being consistent with experiments, the created skyrmions can be retained (Fig. 2g)". I do not agree that the simulations are consistent with experiment. The experiment (Fig. 2) does not show retention of created skyrmions (it just shows retention of a skyrmion state) and is thus not consistent with simulation which shows that the individual skyrmions can be retained.

For simulations in Fig. 2, the authors take magnetoleastic anisotropy modification and additionally variations of DMI with electric field into account (see discussion above). Have the authors taken the same values of DMI vs electric field also for simulations in Figs. 4 and 5 or do they only assume a magnetoelastic change of anisotropy in these simulations? Can the simulation reproduce the reappearance of a skyrmion in Fig. 5c if the authors assume/disregard change of DMI with strain?

I also have some concerns regarding the extraction of DMI from BLS. When using the BLS technique to determine DMI, how do the authors exclude possible non-reciprocal effects due to dipolar interactions, which can be very pronounced in multilayers (e.g. PHYSICAL REVIEW APPLIED 12, 034012 (2019))? In particular, any strain-based modification of anisotropy could lead to a change of nonreciprocity (unrelated to DMI). According to Eq. (8) in this PRA, such a nonreciprocity also increases with increasing k . Is Eq. (1), which applies for the spin wave dispersion of a single magnetic layer, still valid for the multilayer and how are spin waves modes in the multilayer treated (see reference above and Phys. Rev. B 41, 530 (1990))?

Minor issues/questions:

- There are several grammatical errors, the manuscript will need further proofreading/editing to improve readability.
- The arrangement of figures and panels is not consistent with the order of their discussion in the text, which makes the manuscript unnecessarily hard to follow.
- The interlayers (4nm Pt + 1.9nm Ta) are probably too thick to allow exchange coupling between the individual Co layers. Is the magnetization uniform along the film normal throughout the individual Co layers?

Overall, I found that the manuscript deals with an interesting and important topic, but I found the experimental data and the manuscript text not very convincing. In particular, I am not convinced that the manuscript provides unambiguous evidence for DMI variation via strain-mediated magnetoelectric coupling as claimed in the conclusion. I thus cannot recommend publication of the manuscript in its present form.

Reviewer #3 (Remarks to the Author):

The authors report on the experimental demonstration of electric field induced skyrmion creation, deformation, and annihilation. It is claimed that this is achieved via strain mediated magneto-electric coupling in a ferromagnetic/ferroelectric heterostructure. This is an interesting result combining two technologically relevant fields of spintronics : (i) electric field control of magnetism and (ii) manipulation of chiral magnetic skyrmions.

However, I question whether there is enough experimental evidence that proves the main point of this report, which is that the strain induced magnetic property variation enables skyrmion manipulation. The major issues and comments are the following:

1. In page 8 (line 153), it is stated that the DMI in this system varies between 0.82 mJ/m^2 and 0.63 mJ/m^2 depending on the applied electric field. This is much smaller than the typically reported DMI in multilayer systems ($\sim 1.5 \text{ mJ/m}^2$) or B20 systems ($\sim 3 \text{ mJ/m}^2$). Can the authors explain how magnetic skyrmions can form with such a small DMI? Moreover, can the authors provide any experimental proof, via direct imaging (MFM or L-TEM) or indirect methods (asymmetric domain expansion under in-plane field), that the skyrmions in this study actually have left-handed chirality as stated in page 11 (line 229-230)?

2. The mechanism of skyrmion creation by electric field induced strain is not so clear. Although both the change in DMI and change in the effective magnetic anisotropy is mentioned in the manuscript, the main mechanism for the skyrmion creation is not clearly stated. As the authors should well know, magnetic domain formation depends on many competing energies, including the DMI, magnetic anisotropy, and magnetic dipole interaction. In earlier studies (Nat. Nanotech. 12, 1040 (2017); Nat. Electronics 1, 288 (2018)), it was reported that the magnetic anisotropy plays an important role in skyrmion generation. Thus, experimental measurement of the voltage induced modulation of the magnetic anisotropy in this system should be provided. Such data would greatly assist in determining the mechanism of voltage induced skyrmion creation.

3. While the authors sell this method on the potential of low power control of skyrmions, they leave out discussion on an important issue with ferroelectric material based devices: the ferroelectric polarization fatigue which limits the stability and durability of the devices. Data that shows the repeatability and reversibility of this device/method should be provided to prove that this method is

feasible in practical devices.

Because of the above raised concerns, I hesitate to recommend this paper for publication in Nature Communications without a major revision based on extended data.

As a minor note not related to the science in this report, I strongly suggest editing help from someone with full professional proficiency in English.

Response to the reviewers' comments

We thank all the reviewers for the constructive and valuable comments on our original manuscript, which helped us greatly to improve our manuscript. In response, we have revised the manuscript thoroughly, and believe that these changes have improved our paper substantially. Below, please find our point-by-point responses to the reviewers' comments as delineated in blue. A summary of main changes has been listed at the end of the response.

Reviewer #1 (Remarks to the Author):

Authors demonstrated electric-field control of skyrmion through strain-mediated magnetoelectric coupling in multiferroic heterostructure. They showed the process of skyrmions creation, isolated skyrmion deformation and annihilation by performing magnetic force microscopy with in situ electric fields. In the present research work the authors confirmed the decisive role of magnetoelastic coupling in skyrmion manipulation through micromagnetic simulations. Although the fact that there are several scientific simulation works on creating, deleting and driving skyrmions the present work is the first experimental work in FM/FE multiferroic heterostructure that is being reported so far providing so a platform for the research of electric-field control of skyrmion in multiferroic heterostructure.

The paper is well organized and the presentation of the results is clear. The conclusions are soundly supported by data presented in the manuscript and in the supplementary data. The paper is very interesting, well written and provides concise explanations, justifications and physical insights towards new directions regarding electric-field manipulation of skyrmion in multiferroic heterostructure via magnetoelectric coupling.

I recommend this paper for publication in "Nature Communications" subject to a number of minor changes and corrections that could be addressed. In order to improve readability I suggest the following:

Response: We thank the reviewer for the very positive comments and recommending the publication of our paper in Nature Communications.

1. The authors should try to include some figures (e.g. Fig. 1: Schematic of the sample configuration, Supplementary Fig. 18: A series of skyrmion morphology under different in-plane biaxial tensile strain.) in order to increase the readability of the paper.

Response: Thank the reviewer for this suggestion. We have added "Schematic of the sample configuration" as Fig. 1a. We also improved the micromagnetics simulations by considering both DMI and the effect of pinning on the disappearance/reappearance of one skyrmion rather than the tensile strains (previous Supplementary Fig. 18) and the results are shown in the insets of Fig. 5.

2. The present study and the experimental procedure took place around room temperature. The authors in their micromagnetics simulations presented in Methods Section they refer to the use of “mumax 3 (version: 3.10 β , which incorporates the module of magnetoelastic coupling) for simulating the evolution of local magnetization in polycrystalline [Pt (4 nm)/Co (1.6 nm)/Ta (1.9 nm)] \times 5 multilayers under voltage-induced strains transmitted from the underlying piezoelectric PMN-PT substrate”. It would be nice for the readability and the reproducibility of the paper to include the actual energy functional used having contributions for instance from exchange, anisotropy, demagnetization and Dzyaloshinskii-Moriya or from thermal fields if we refer to temperatures differing from 0K. Presumably (unfortunately is not clear in the manuscript), the micromagnetic simulations took place at 0K. The authors should comment on this in a more detailed manner since they compare results from simulations at 0K with experimental measurements at elevated temperatures.

Response: We thank the reviewer for the suggestions. In the revised paper, we have made a few changes (please see highlighted text in Methods section and the Supplementary Figure S19). These changes are briefly summarized below.

- All relevant energy functionals are now included in the revised Methods section.
- In revised Methods section, we clarify that the temperature fluctuation field is set as 0 (temperature $T=0$ K) for the simulation results shown in the main text, and that the reason for that is to perform a cleaner analysis on the energetics of skyrmion switching.
- Influence of thermal fluctuation at room temperature (298 K) on the skyrmion switching is presented in Supplementary Figure S19.

A comparison of the simulation results obtained under zero (0 K) and room-temperature (298 K) thermal fluctuation field is also shown below. Note that these simulation results were performed using the newly measured interfacial DMI strength (which was suggested by Reviewer #2).

Figure R1 | A comparison of the micromagnetic simulation results at 0 K and 298 K. with/without thermal fluctuation. Although “0 K” and “298 K” are used, all materials

parameters used for micromagnetic simulations are measured at room temperature, based on either our own experiments or others' experiments reported in literature.

As seen, the overall behavior of the strain-mediated creation of skyrmion under $T=298$ K is similar to both the simulation results obtained under $T=0$ K, hence being similar to experimental observations (see Fig. 3a-d in the main paper). Regarding the details of the switching process, two observations were made. First, the addition of a room-temperature (298 K) thermal fluctuation field introduces white noise into the magnetization distribution. Second, compared to the case without thermal fluctuation field (0 K), there are more skyrmions created when in-plane compressive strain is applied with $E = -4$ kV/cm, and more skyrmions retained when the electric field is turned off ($E=0$ kV/cm).

Reviewer #2 (Remarks to the Author):

1. The authors show experimental indications for creation, deformation and annihilation of skyrmions in ferromagnetic multilayers by uniform elastic strain. The strain is caused by application of a voltage to piezoelectric PMN-PT substrates, such that an indirect electric field control of the above processes is achieved.

Response: Thanks for this comment.

The authors explain their experimental observations by assuming that the strain changes both, magnetic anisotropy and DMI. I found the extraction of the DMI-modification by electric fields, which is then used as input for micromagnetic simulations, unconvincing. My detailed reasoning is the following:

As shown in Fig. 3a, electric-field control of strain is hysteretic and quite unusual in their particular sample (same sign of strain for positive/negative electric fields, large asymmetry). In Fig. 3a, Fig. 2 and Fig. 4, the authors use electric fields in the range ± 4 kV/cm. However, in Fig. 3b, the authors use ± 8 kV/cm. The authors then assume that strain, electric field and DMI are all linearly related to extract DMI at ± 4 kV/cm from Fig. 3b, which is then used in their micromagnetic simulations. This leads to the following inconsistency:

Fig. 3a shows that positive/negative electric fields lead to drastically different strains. For $E > 0$, the strain is very small (factor 10 smaller than for negative E). However, in Fig. 3b, for $E = \pm 8$ kV/cm, similar DMI is found. From this figure the authors then extract similar DMI for ± 4 kV/cm (0.73 and 0.76 mJ/m²) by linear interpolation. The finding of such similar values of DMI for different E polarities is inconsistent with a strain-based origin of the DMI modification, considering that Fig. 3a demonstrates very different strain for ± 4 kV/cm. **Notably, in Fig. 4d the authors observe no skyrmion deformation with $E > 0$, consistent with Fig. 3a but inconsistent with Fig. 3b. (We reply to this comment in Point 2)**

The authors need to perform the DMI extraction experiments using the same electric field hysteresis loops (i.e. from +4kV/cm to -4kV/cm and not from +8kV/cm to -8kV/cm) as used for the rest of the study if they want to extract meaningful DMI values as input for their micromagnetic simulations. With the data present in the manuscript, the assumption that the DMI is essentially unchanged under electric field polarity reversal (Fig. 3b) is inconsistent with their further observations.

Response: Thanks for the suggestion — “perform the DMI extraction experiments using the same electric field hysteresis loops (i.e. from +4kV/cm to -4kV/cm and not from +8kV/cm to -8kV/cm) as used for the rest of the study”.

Following this suggestion, we performed the DMI extraction experiments using the electric-field hysteresis loops (from +4 kV/cm to -4 kV/cm) and the results are shown in Fig. R2. The inset of Fig. R2 shows D of 0.73 mJ/m^2 for +4 kV/cm, 0.77 mJ/m^2 for +0 kV/cm, and 0.58 mJ/m^2 for -4 kV/cm, which indicate a small difference in DMI values between +0 kV/cm and +4 kV/cm, and a remarkable difference between +0 kV/cm and -4 kV/cm. This behavior is consistent with the strain curve shown in Fig. R3 (previous Fig. 3a, updated Fig. 2a in the main text of the manuscript) and indicates electric-field-induced DMI variation via strain-mediated magnetoelectric coupling. Moreover, this is also consistent with our experimental observations that there is significant manipulation on skyrmions under electric-field of -4 kV/cm, and no significant manipulation on skyrmions under electric-field of +4 kV/cm.

We then used the DMI values of +4 kV/cm to -4 kV/cm loop for updating micromagnetic simulation. The obtained results are consistent with our experimental results, as shown in Fig. 3 (e)-(h) of the main text of the revised manuscript.

In order to exclude the possible influence of electric-field polarity on the DMI measurements, we also measured the DMI of -1.4 kV/cm (+4 kV/cm to -4 kV/cm branch) in addition to +4 kV/cm because its strain is comparable to that of +4 kV/cm according to Fig. R3 (+4 kV/cm to -4 kV/cm branch, Fig. 2a in the main text of the revised manuscript). It can be seen from the inset of Fig. R2 that the DMI 0.71 mJ/m^2 for -1.4 kV/cm is comparable to the DMI value (0.73 mJ/m^2) of +4 kV/cm, excluding the possible influence of electric-field polarity on the DMI measurements.

Figure R2 | Wave-vector dependence of Δf under different electric fields. The inset shows the DMI values with the error bar obtained from the standard error of Lorentzian fitting.

Figure R3 | Out-of-plane strain variation with electric field for PMN-PT(001) substrate.

We replaced Fig. 2b with Fig. R2 in the revised manuscript (page 35).

2. Notably, in Fig. 4d the authors observe no skyrmion deformation with $E > 0$, consistent with Fig. 3a but inconsistent with Fig. 3b.

Response: As mentioned in Point 1, we followed the reviewer's suggestion to perform

the DMI extraction experiments using the same electric-field hysteresis loops (from +4 kV/cm to -4 kV/cm). The results can account for the reviewer's question. According to the inset of Fig. R2, the DMI value is 0.73 mJ/m² for +4 kV/cm, which is close to the DMI value at +0 kV/cm (0.77 mJ/m²). In contrast, at -4 kV/cm, the DMI value is 0.58 mJ/m², which is significantly different from the DMI value at +0 kV/cm. In general, the DMI values for +4 kV/cm and -4kV/cm differ greatly, and the variation trend of DMI value is consistent with that of the strain. Therefore, no skyrmion deformation observed with E>0 is consistent with Fig. R2, which replaces the previous Fig. 3b (Fig. 2b in the revised manuscript).

3. Furthermore, I also found the data in Fig. 2 somewhat inconclusive. First, the authors should clarify how they made sure that Fig. 2 a-d all show the exact same location on the sample. Second, in Fig. 2, there does not appear to be a clear correlation between the domains/skyrmions in panels b,c,d. This is different to the micromagnetic simulations, where one can identify that a large number of skyrmions are unchanged between panels g and h. If the sample stays in a skyrmion state when switching off the voltage but individual skyrmions are not retained, this does not seem to be very useful for memory applications and it also seems to contrast the findings in Fig. 5. Also, the authors state on page 9, line 190 "Being consistent with experiments, the created skyrmions can be retained (Fig. 2g)". I do not agree that the simulations are consistent with experiment. The experiment (Fig. 2) does not show retention of created skyrmions (it just shows retention of a skyrmion state) and is thus not consistent with simulation which shows that the individual skyrmions can be retained.

Response: These comments contain three points and we reply to them one by one.

a. First, the authors should clarify how they made sure that Fig. 2 a-d all show the exact same location on the sample.

Response: The MFM images were taken in the tapping/lift mode, i.e. the topography and magnetic images were obtained at the same time. The topography of the sample is obtained using the tapping mode for the first scan, and the magnetic image is obtained using the lift mode for the second scan. The magnetic images in previous Fig. 2a-d (Fig. 3a-d in the revised manuscript) and their corresponding topography images are shown in Fig. R4. The similar topography images indicate that previous Fig. 2 a-d (Fig. 3a-d in the revised manuscript) show the exact same location on the sample.

Fig. R4| MFM images at $E = +0$ kV/cm (a), -4 kV/cm (b), -0 kV/cm (c), $+4$ kV/cm (d) with $B_{\text{bias}} = 60$ mT and the corresponding topography images (e)-(h).

We added above discussion and Fig. R4 in the revised Supplementary Information-Supplementary Figure 12 (page 11).

b. Second, in Fig. 2, there does not appear to be a clear correlation between the domains/skyrmions in panels b,c,d. This is different to the micromagnetic simulations, where one can identify that a large number of skyrmions are unchanged between panels g and h.

Response: The correlation between the domains/skyrmions in panels b, c, d is as follows. In panel b, skyrmions appear under -4 kV/cm. Most skyrmions remain after removing -4 kV/cm as shown in panel c (Fig. R5). The change of skyrmion state is minor under $+4$ kV/cm in panel d (Fig. R6). In fact, a large number of skyrmions are unchanged between panels c and d, which is similar to the behavior in the corresponding micromagnetic simulations (panels g and h).

Fig. R5| MFM images at $E = -0$ kV/cm (a), $+4$ kV/cm (b), with $B_{\text{bias}} = 60$ mT.

Fig. R6 | MFM images at $E = -4$ kV/cm (a), -0 kV/cm (b), with $B_{\text{bias}} = 60$ mT.

c. If the sample stays in a skyrmion state when switching off the voltage but individual skyrmions are not retained, this does not seem to be very useful for memory applications and it also seems to contrast the findings in Fig. 5.

Response: For the samples of continuous films rather than patterned nanostructures, it's difficult to make every skyrmion show the same behavior considering the inhomogeneity related to the defects (grain boundary, impurity, etc.), which is also important for skyrmions and is difficult to deal with. Actually, this is a challenge in this field and deserves further study as mentioned in the very recent review (J. Phys. D: Appl. Phys. 53, 363001 (2020)). In our paper, the scan ranges for Fig. 2 ($5 \mu\text{m}$) and Fig. 5 ($2 \mu\text{m}$) are different, with the former showing the behavior of many skyrmions and the latter showing the behavior of one skyrmion. Therefore, the distinguishability of an individual skyrmion in Fig. 5 is better than that in previous Fig. 2 (Fig. 3 in the revised manuscript), which is good for revealing the deformation of one skyrmion or the change of one skyrmion. So the purposes of Fig. 2 (Fig. 3 in the revised manuscript) and Fig. 5 are different, and Fig. 5 shows the behavior of one skyrmion, which is different from others. The key point is that the deposited multilayers are polycrystalline, in which defects cannot be ignored. "Defects result in skyrmion pinning, and strongly affect skyrmion motion and also skyrmion creation and destruction." as mentioned in a very recent review (J. Phys. D: Appl. Phys. 53, 363001 (2020)). It should be mentioned that the stray field of MFM magnetic tip inevitably interacts with skyrmions and distorts them (Appl. Phys. Lett. 112, 132405 (2018)), which may also affect the behavior of some individual skyrmions although our control experiment (Supplementary Figure. S10) shows that such weak stray field from the MFM tip cannot lead to skyrmion creation

Thus, the main significance of our work is the observation of electric-field control of skyrmion through strain-mediated magnetoelectric coupling in multiferroic thin-film heterostructures for the first time. While for the memory applications of skyrmions, the retention of an individual skyrmion or every skyrmion needs more effort in the future, notably on harnessing defects-related effects for a more precise control of the

skyrmions. This issue is still a challenge in this field. Despite this, there are some ways for controlling individual skyrmion in nanostructures. For example, strain-mediated voltage-controlled switching of an isolated magnetic skyrmion in nanostructures has been theoretically demonstrated (npj Computational Materials. 4, 62, 2018) and can potentially be used to design skyrmions-based low-power magnetic random-access memory.

d. Also, the authors state on page 9, line 190 “Being consistent with experiments, the created skyrmions can be retained (Fig. 2g)”. I do not agree that the simulations are consistent with experiment. The experiment (Fig. 2) does not show retention of created skyrmions (it just shows retention of a skyrmion state) and is thus not consistent with simulation which shows that the individual skyrmions can be retained.

Response: Please see the response to question b.

4. For simulations in Fig. 2, the authors take magnetoelastic anisotropy modification and additionally variations of DMI with electric field into account (see discussion above). Have the authors taken the same values of DMI vs electric field also for simulations in Figs. 4 and 5 or do they only assume a magnetoelastic change of anisotropy in these simulations?

Can the simulation reproduce the reappearance of a skyrmion in Fig. 5c if the authors assume/disregard change of DMI with strain?

Response: In the previous simulations for Figs. 4 and 5, we did not take the change of DMI with strain into consideration by assuming the dominant role of magnetoelastic change of anisotropy in these two cases. Following the reviewer’s suggestion, we considered strain-induced change in the DMI and updated the simulations and relevant discussion in the main text. The updated simulations and discussions provide a better interpretation of the mechanisms for both the reversible deformation and deletion/reappearance of one single skyrmion. Details are as follows.

a. On the reversible deformation of the skyrmion (see Fig. R7 below and updated Fig. 4)

As shown in Fig. R7a-b, when electric field changes from +0 kV/cm to -4 kV/cm, the interfacial DMI strength (D) of the entire multilayer, as measured by BLS, decreases from 0.7721 mJ/m² to 0.5850 mJ/m², which is induced by the tensile average out-of-plane strains measured by XRD. However, concerning the behaviors of a single skyrmion in a local area, what really matter are the local strain and the associated local D . It has been shown that the 109° ferroelectric domain switching of the (001) PMN-PT single crystal can lead to locally anisotropic strains along the [-110] and [110] crystal axes. As a result, the local DMI along these two axes can also be anisotropic. But unfortunately, direct experimental measurement of the local strain and the local D in

such multiferroic heterostructures still remain to be a challenge.

Figure R7 | The simulation results of strain-mediated deformation of one single skyrmion with,

- (a) $\varepsilon_{[-110]} = \varepsilon_{[110]} = 0$; $D = 0.7721 \text{ mJ/m}^2$ for $+0 \text{ kV/cm}$,
 - (b) $\varepsilon_{[-110]} = -0.169\%$, $\varepsilon_{[110]} = 0$; $D = 0.5850 \text{ mJ/m}^2$ for -4 kV/cm
 - (c) $\varepsilon_{[-110]} = \varepsilon_{[110]} = -0.023\%$; $D = 0.6852 \text{ mJ/m}^2$ for -0 kV/cm
 - (d) $\varepsilon_{[-110]} = \varepsilon_{[110]} = -0.012\%$; $D = 0.7267 \text{ mJ/m}^2$ for $+4 \text{ kV/cm}$
- The scale bar is 100 nm. The bias magnetic field $B_{\text{bias}} = 30 \text{ mT}$.

In view of these, in the revised paper, we first outlined a few possible mechanisms of the observed skyrmion deformation: it could be caused by the locally anisotropic strain (Acta Materialia 183, 145 (2020)), the locally anisotropic DMI caused by the anisotropic strain (NATURE NANOTECHNOLOGY 10, 589 (2015)) only, or both. Then, micromagnetic simulations were performed to demonstrate one of the many possible conditions that can lead to the deformation of a single skyrmion to the extent observed by experiments. As shown in Fig. 7Rb, under a reasonable in-plane compressive uniaxial strain ($\varepsilon_{[-110]} = -0.169\%$, $\varepsilon_{[110]} = 0$) and an isotropic D of 0.5850 mJ/m^2 , a deformation of $\sim 47\%$ (similar to experiment) is obtained. Because D is isotropic, the deformation is purely caused by the anisotropic strain. Yet again, the same simulation results can be obtained by using an anisotropic D only. Regardless of these details, the deformation of a skyrmion results from the imbalance of the Néel wall energy σ_w , which is related to both the local anisotropy and the local D (Acta Materialia 2020, 183, 145). Specifically, the skyrmion radius is smaller along the axis with a lower wall energy σ_w . This is analogous to Wulff construction: lower surface energy (wall energy) of a crystal (skyrmion) yields shorter vector length (skyrmion radius) at thermodynamic equilibrium.

We have added the discussion above to the revised main text (page 14-16) and replaced relevant figures in the original Fig. 4 with Fig. R7.

b. On the re-appearance of skyrmion (Fig. R8 below, and the updated Fig. 5)

Regarding the simulation to reproduce the reappearance of a skyrmion, we found that the deleted skyrmion won't reappear no matter the strain-induced change of DMI is incorporated or disregarded. Figure R8a-d show the simulation results with electric-field-induced strain and change of D being incorporated. For simplicity, average strain

and average D measured by our experiments are considered. As seen, when electric field changes from +0 kV/cm to -4 kV/cm, the D reduces from 0.7721 mJ/m² to 0.5850 mJ/m². The reduction in D is the sole reason that leads to the deletion of skyrmion (from Fig. R8a to Fig. R8b), with no contribution from the strain. This is because the biaxial in-plane compressive strain ($\varepsilon_{[-110]} = \varepsilon_{[110]} = -0.0425\%$) at -4 kV/cm, albeit small, would lead to the expansion of skyrmion in Co with negative magnetostriction. Once the skyrmion is fully deleted, it won't re-appear even when the change of D is incorporated, as shown in Fig. R8c-d.

Figure R8 | The simulated “annihilation” and reappearance of one single skyrmion with,
(a)(e) $\varepsilon_{[-110]} = \varepsilon_{[110]} = 0$, $D = 0.7721$ mJ/m² for $E = +0$ kV/cm,
(b)(f) $\varepsilon_{[-110]} = \varepsilon_{[110]} = -0.0425\%$, $D = 0.5850$ mJ/m² for $E = -4$ kV/cm,
(c)(g) $\varepsilon_{[-110]} = \varepsilon_{[110]} = -0.0415\%$, $D = 0.6852$ mJ/m² for $E = -0$ kV/cm.
(d)(h) $\varepsilon_{[-110]} = \varepsilon_{[110]} = -0.012\%$, $D = 0.7267$ mJ/m² for $E = +4$ kV/cm.
(a-d): a 20-nm-diameter pinning site with $\sim 5\%$ lower perpendicular anisotropy was specified, as indicated by the dashed circle in the center.
(e-h): no pinning site: the perpendicular anisotropy is spatially uniform.
The scale bar is 100 nm. The bias magnetic field $B_{\text{bias}} = 30$ mT.

The reason why skyrmion does re-appear experimentally may be attributed to defects or pinning sites. The role of pinning site in the reappearance of skyrmions was shown by Bhattacharya et al. (NATURE ELECTRONICS 3, 539 (2020)) in the electric field gating experiment, skyrmions only reappeared in the regions with pinning sites, and they mentioned that “Finally, some skyrmions were created at the same initial location that they occupied before annihilation. We found that these locations are the low-anisotropy regions of the film, which act as pinning sites for the skyrmions.”

Inspired by this work, we introduced a 20-nm-diameter pinning site with $\sim 5\%$ lower perpendicular anisotropy at the center of the simulation system, and re-do the simulations under the same conditions of strain and D . The simulation results are shown

in Figs. R8e-h for comparison. As shown, this lower-anisotropy pinning site leads to the stabilization of a tiny (diameter ~ 10 nm) skyrmion at -4 kV/cm (see Fig. R8f), which is too small to discern with our MFM (with a spatial resolution of ~ 10 nm as well), yet can function as the nucleus for the re-appearance (growth) of skyrmion at exactly the same location after the removal of electric field (Fig. R8g). Without nucleus as such, the skyrmion may not necessarily re-appear from exactly the same location where it annihilated.

Overall, these new simulation results not only provide a reasonable interpretation for the experimentally observed annihilation and re-appearance of a single skyrmion from the same location of a continuous magnetic layer, but also point out the possibility of harnessing pinning sites for realizing a spatially precise skyrmion manipulation that could be useful for device applications.

We have added the discussion above to the main text (page 16-17) and replaced relevant figures in the original Fig. 5 with Fig. R8e-f.

5. I also have some concerns regarding the extraction of DMI from BLS. When using the BLS technique to determine DMI, how do the authors exclude possible non-reciprocal effects due to dipolar interactions, which can be very pronounced in multilayers (e.g. PHYSICAL REVIEW APPLIED 12, 034012 (2019))? In particular, any strain-based modification of anisotropy could lead to a change of nonreciprocity (unrelated to DMI). According to Eq. (8) in this PRA, such a nonreciprocity also increases with increasing k . Is Eq. (1), which applies for the spin wave dispersion of a single magnetic layer, still valid for the multilayer and how are spin waves modes in the multilayer treated (see reference above and Phys. Rev. B 41, 530 (1990))?

Response: The reviewer comments consist three questions and we address them one by one as follows.

a. When using the BLS technique to determine DMI, how do the authors exclude possible non-reciprocal effects due to dipolar interactions, which can be very pronounced in multilayers (e.g. PHYSICAL REVIEW APPLIED 12, 034012 (2019))?

Response to question a: The non-reciprocal effect due to the dipolar interaction is indeed pronounced in multilayers, however, it can be ignored for our work because the conditions for it to be effective are not satisfied. The conditions are the magnetic layers are antiferromagnetically coupled, or ferromagnetically coupled with the magnetic properties (such as M_s) of the magnetic layers different. These conditions are required for the dipolar interactions in the multilayer film to give rise to the non-reciprocal effects (this theory proposed by Grunberg, JOURNAL OF APPLIED PHYSICS 52, 6824 (1981). For our work, each magnetic layer has the same material and thickness, and we applied the in-plane saturation magnetic field of 5000 Oe during the measurement of BLS, so that the magnetic moments of the magnetic layers are parallel.

In order to check whether the magnetic properties of each layer are equivalent, we grew Si/Ta(4.7)/[Pt(4)/Co(1.6)/Ta(1.9)]_n multilayers with different periods (n is 1, 3 and 5, respectively) by magnetic sputtering. It is found that their in-plane saturation magnetic moments are proportional to the number of period as shown in Fig. R9. As a result, it can be determined that the magnetic properties of each layer of our sample are equivalent. So it can be concluded that conditions for the dipolar interaction-induced non-reciprocal effect to be effective are not satisfied for our sample. For these reasons, the non-reciprocal effects due to dipolar interactions in our sample are negligible. Moreover, even if there is a weak non-reciprocal effect due to the dipolar interactions, considering that the sample's non-magnetic layer thickness is 5.9 nm, the non-reciprocal effects due to dipolar interactions decreases exponentially with the increase of the non-magnetic layer thickness (PHYSICAL REVIEW APPLIED 12, 034012 (2019)), so these non-reciprocal effects can be ignored in our work.

Figure R9 | In-plane saturation magnetic moments of Si/Ta(4.7)/[Pt(4)/Co(1.6)/Ta(1.9)]_n (n is 1, 3 and 5).

To check this directly by experiment, we grew Si/Ta(4.7)/[Pt(4)/Co(1.6)/Ta(1.9)]_n (n is 1, 3 and 5) by magnetic sputtering to measure their DMI values by BLS. Within the error range, the DMI values of the three samples are equivalent as shown in Fig. R10. This independence of DMI on the number of magnetic layer (from single layer to multilayers) indicates that the dipolar interactions are not effective in our samples.

Figure R10 | Wave-vector dependence of Δf in Si/Ta(4.7)/[Pt(4)/Co(1.6)/Ta(1.9)]_n (n is 1, 3 and 5) with the DMI values in the inset with the error bar obtained from the standard error of Lorentzian fitting.

We added above discussion and Fig. R9 & R10 in the revised Supplementary Information-Supplementary Figure 8 (page 7).

b. In particular, any strain-based modification of anisotropy could lead to a change of nonreciprocity (unrelated to DMI).

Response to question b: Strain indeed affects the magnetic anisotropy and thus modify the spin wave frequency. However, it does not change the nonreciprocity of frequency. The reason is as follows. To account for various contributions, the spin wave dispersion relation is described by the following formula (PHYSICAL REVIEW LETTERS 114, 047201 (2015)).

$$\omega = \omega_0 + \omega_{DM}$$

$$\omega_0 = \mu_0 \gamma \sqrt{[H_0 + Jk^2 + \xi(kL)M_S][H_0 - H_U + Jk^2 + M_S - \xi(kL)M_S]}$$

$$\omega_{DM} = -\frac{2\gamma}{M_S} Dk$$

where H_U is the effective field of magnetic anisotropy. From this, it can be deduced that magnetic anisotropy indeed affects the spin wave frequency. However, we use $\omega(-k) - \omega(k)$ (nonreciprocity of frequency) to get the DMI value. The contribution of H_U to ω_0 does not depend on the sign of k , which also apply to other terms in ω_0 in addition to H_U . As a result, the frequency difference of counterpropagating spin waves, given by

$$\Delta f(k) = \frac{[\omega(-k) - \omega(k)]}{2\pi} = \frac{2\gamma}{\pi M_S} Dk \quad (\text{PHYSICAL REVIEW LETTERS 114, 047201})$$

(2015)), is only determined by the DMI interaction. So the strain-based modification of anisotropy affects the spin wave frequency, but it does not change the nonreciprocity of frequency. This means that it doesn't affect the measurement of DMI which is obtained by subtracting the frequencies for the positive and negative wave vector, based on DMI-induced nonreciprocity of frequency.

c. According to Eq. (8) in this PRA, such a nonreciprocity also increases with increasing k . Is Eq. (1), which applies for the spin wave dispersion of a single magnetic layer, still valid for the multilayer and how are spin waves modes in the multilayer treated (see reference above and Phys. Rev. B 41, 530 (1990))?

Response to question c: According to the reference (PHYSICAL REVIEW APPLIED 12, 034012 (2019)), the nonreciprocity increases with increasing k by dipolar interaction in the bilayers whose magnetizations are different. However, as described in the “**Response to question a**”, the saturation magnetization M_s of the magnetic layers are equivalent in our work, and each Co layer's thickness is equal, so the non-reciprocal effects of the dipolar interactions can be ignored. As a result, Eq. (1) is still valid in our work. As mentioned in **Response to question a Fig. R10**, we grew Si/Ta(4.7)/[Pt(4)/Co(1.6)/Ta(1.9)] $_n$ (n is 1, 3 and 5) by magnetron sputtering to measure their DMI values and the DMI values were equivalent as shown in Fig. R10, which corroborates our theory.

For the problem of spin waves modes in the multilayer, according to the reference (PHYSICAL REVIEW B 41, 530 (1990)), it mainly discussed the increase in the number of modes for the vertical standing spin-wave as the thickness increases. The standing spin-wave modes coupled with the surface spin-wave mode, and the surface spin-wave mode is still one. Only the surface spin-wave mode determines the DMI. In other words, the standing spin-wave modes don't affect the BLS measurement.

In the part V. MAGNETIC/NONMAGNETIC MULTILAYERS of the paper (PHYSICAL REVIEW B 41, 530 (1990)) mentioned by the referee, there are some statements that “For a single magnetic layer there exists one dipolar surface mode nearly unaffected by exchange”, and “For a magnetic/nonmagnetic multilayer the role of volume exchange interaction is replaced by interlayer exchange. For large spacer thicknesses, the latter can be neglected and we are left with the purely dipolar coupled modes”. So, according to these statements, the thickness of the space layer is important. For our samples, the space layers (4 nm Pt + 1.9nm Ta) are 5.9 nm thick, so the interlayer exchange can be ignored, as the reviewer pointed out in the next question. There is no optical branch, and each layer should be independent. Because each magnetic layer has the same properties, only the superposition of single layer signals can be obtained in the experiment.

6. Minor issues/questions: There are several grammatical errors, the manuscript will need further proofreading/editing to improve readability.

Response: The revised manuscript was proofread by a professor in the USA and the grammatical errors should be fixed.

7. The arrangement of figures and panels is not consistent with the order of their discussion in the text, which makes the manuscript unnecessarily hard to follow.

Response: We have adjusted the arrangement of Fig.2 & 3 to make them consistent with the order of their discussion in the main text of the revised manuscript.

8. The interlayers (4nm Pt + 1.9nm Ta) are probably too thick to allow exchange coupling between the individual Co layers. Is the magnetization uniform along the film normal throughout the individual Co layers?

Response: Indeed, the spacer or interlayers are too thick to allow exchange coupling between the individual Co layers. We have studied the vertical magnetization distribution within a single Co layer of the sample by micromagnetic simulation as shown in Fig. R11. The micromagnetic simulations were performed based on a multilayer system, where the magnetic Co layers are separated by non-magnetic spacers. Both the Co layers and the non-magnetic layers in the simulations have almost the same thickness as those in experiment. It can be seen that the magnetization of the first and second layer is always uniform along the film normal throughout the individual Co layers.

Figure R11 | Magnetization distribution of two Co atomic layers of the individual Co layer. The magnetization distribution in the first Co atomic layer is shown in Fig.

R9a~d for $E=+0$ kV/cm (a), -4 kV/cm (b), 0 kV/cm (c), $+4$ kV/cm (d). The magnetization distribution in the second Co atomic layer is shown in Fig. R9e~h for $E=+0$ kV/cm (e), -4 kV/cm (f), 0 kV/cm (g), $+4$ kV/cm (h).

9. Overall, I found that the manuscript deals with an interesting and important topic,

but I found the experimental data and the manuscript text not very convincing. In particular, I am not convinced that the manuscript provides unambiguous evidence for DMI variation via strain-mediated magnetoelectric coupling as claimed in the conclusion. I thus cannot recommend publication of the manuscript in its present form.

Response: We appreciate the reviewer's comment on our paper as "the manuscript deals with an interesting and important topic". First, we ruled out the interference of dipolar interaction, magnetic anisotropy and possible electric-field polarity on the BLS measurement. Second, the DMI values were measured for the +4 kV/cm to -4 kV/cm loop. The DMI value for +4 kV/cm differs greatly from the DMI value for -4 kV/cm, and this difference is consistent with the strains for +4 kV/cm and -4 kV/cm. The change of DMI with electric-field is consistent with the change of the strain with electric-field, indicating strain-induced change of DMI. Moreover, for our work, the electric field gating effect can be ignored due to the short charge-screening length in metallic ferromagnetic layer.

In general, it is strain-mediated modulation of the Fert-Levy DMI at the HM/FM (Pt/Co) interface in our work. Tensile strain transferred from PMN-PT increases the distance between Co and Pt atom at the interface and thereby reduces the strength of hybridization, leading to weakening of interfacial DMI. Therefore, the DMI variation via strain-mediated magnetoelectric coupling is demonstrated in our work.

In conclusion, we report magnetic skyrmions in the FM/FE multiferroic heterostructure, electric-field control of the Fert-Levy DMI and magnetic anisotropy via strain-mediated magnetoelectric coupling. We experimentally observe electric-field manipulation of skyrmions, especially creation, deformation and annihilation. Micromagnetic simulation indicated the importance of magnetoelastic coupling in these processes.

Reviewer #3(Remarks to the Author)

The authors report on the experimental demonstration of electric field induced skyrmion creation, deformation, and annihilation. It is claimed that this is achieved via strain mediated magneto-electric coupling in a ferromagnetic/ferroelectric heterostructure. This is an interesting result combining two technologically relevant fields of spintronics: (i) electric field control of magnetism and (ii) manipulation of chiral magnetic skyrmions.

However, I question whether there is enough experimental evidence that proves the main point of this report, which is that the strain induced magnetic property variation enables skyrmion manipulation. The major issues and comments are the following.

Response: We appreciate the reviewer's comment on our paper as "This is an interesting result combining two technologically relevant fields of spintronics: (i) electric field control of magnetism and (ii) manipulation of chiral magnetic skyrmions."

We have revised the original manuscript according to the reviewer's comments and provided enough experimental evidence that strain induced magnetic property variation enables skyrmion manipulation.

1. In page 8 (line 153), it is stated that the DMI in this system varies between 0.82 mJ/m² and 0.63 mJ/m² depending on the applied electric field. This is much smaller than the typically reported DMI in multilayer systems (~1.5 mJ/m²) or B20 systems (~3 mJ/m²). Can the authors explain how magnetic skyrmions can form with such a small DMI?

Response: The values of DMI vary remarkably with material systems (Physical Review Letters. 120, 157204 (2018)). BLS is the most direct and standard method to get DMI values for the multilayer systems. To our knowledge, there is only one report on the DMI value of Pt/Co/Ta multilayers (Physical Review B. 100, 144435 (2019)) measured by BLS. They gave DMI value with a magnitude of 0.78 ± 0.02 mJ/m², which is comparable or close to our DMI values of Pt/Co/Ta multilayers measured also by BLS. They also mentioned that "The value of the DMI is sufficient to support the formation of Néel skyrmions", which is also supported by their experimental observation. There is also a report on the values of DMI of Pt/Co/Ta multilayers calculated based on the magnetic domain structure and the domain wall energy for different Co thicknesses, and skyrmions were observed for these samples (ACS. Applied Materials Interfaces. 11, 12098 (2019)) with the values of DMI comparable to our work. Moreover, the simulations in this work also shows that skyrmions can form with these DMI values.

Moreover, can the authors provide any experimental proof, via direct imaging (MFM or L-TEM) or indirect methods (asymmetric domain expansion under in-plane field), that the skyrmions in this study actually have left-handed chirality as stated in page 11 (line 229-230)?

Response: For the chirality of skyrmions in Pt/Co/Ta multilayers, there have been some reports demonstrating left-handed chirality (Communications Physics. 1, 36 (2018); Physical Review B. 100, 144435 (2019); Nat. Materials 15, 501 (2016)).

The chirality of our samples cannot be directly obtained by MFM and L-TEM measurements. MFM is only sensitive to the vertical component of the stray magnetic field of the sample, so it is not an ideal method of obtaining skyrmion chirality. L-TEM can directly obtain the chirality of Bloch-type skyrmions. But samples with PMA must be tilted to observe the Néel-type skyrmions, which makes the identification of the chirality of skyrmions directly from the dark-bright contrast of L-TEM images impermissible (Communications Physics. 1, 36 (2018)). However, as demonstrated by Senfu Zhang et al, observation of the asymmetric domain expansion under an in-plane magnetic field by L-TEM can be used to get the chirality of Néel-type skyrmions (Communications Physics. 1, 36 (2018)). Therefore, based on the reviewer's comments,

we observed asymmetric domain expansion under an in-plane magnetic field by L-TEM. For Pt/Co/Ta multilayers, Senfu Zhang et al showed that “on decreasing the magnetic field, individual skyrmions appear to subsequently evolve into snake-like structures growing in the direction opposite to the in-plane magnetic field”, which illustrate that these skyrmions have left-handed chirality (Communications Physics. 1, 36 (2018)). As shown in Fig. R12, we indicated the changes in the images with the green dashed ellipses. The directions that the snake-like structures preferred to grow along are also opposite to that of the in-plane magnetic field in our work, consistent with that of Senfu Zhang et al’s work (Communications Physics. 1, 36 (2018)). This proves that the skyrmions in our Pt/Co/Ta multilayers actually have left-handed chirality, which has also been demonstrated by the previous reports as mentioned above.

Fig. R12| In-situ L-TEM observation. The images were taken at a tilt angle of $\alpha = 20.06^\circ$ and $\beta = 20.39^\circ$, and the arrow indicates the in-plane magnetic field direction.

We added above discussion and Fig. R12 in the revised Supplementary Information-Supplementary Figure 21 (page 20).

2. The mechanism of skyrmion creation by electric field induced strain is not so clear. Although both the change in DMI and change in the effective magnetic anisotropy is mentioned in the manuscript, the main mechanism for the skyrmion creation is not clearly stated. As the authors should well know, magnetic domain formation depends on many competing energies, including the DMI, magnetic anisotropy, and magnetic dipole interaction. In earlier studies (Nat. Nanotech. 12, 1040 (2017); Nat. Electronics 1, 288 (2018)), it was reported that the magnetic anisotropy plays an important role in skyrmion generation. Thus, experimental measurement of the voltage induced modulation of the magnetic anisotropy in this system should be provided. Such data would greatly assist in determining the mechanism of voltage induced skyrmion creation.

Response: We agree with the reviewer that the magnetic anisotropy plays an important role in skyrmion generation. Based on the reviewer’s comment, we performed angle-dependent ferromagnetic resonance (FMR) measurements. As shown in Fig. R13a, θ is

the angle between the applied magnetic field H and the out of plane (z) direction. For each angle θ , a resonance field $H_r(\theta)$ can be determined from the FMR spectrum. The magnetic anisotropy (denoted as K_{eff}) can be determined by fitting $H_r(\theta)$ with the Kittel formula for FMR, shown in the Supplementary Information S--. The results of angle-dependent FMR measurements and corresponding Kittel formula fitting under different electric fields are shown in Fig. R13b. The magnetic anisotropy under $E = +4, +0, -4, -0 \text{ kV/cm}$, $K_{\text{eff}} = 3.01, 3.39, 0.99, 2.36 \times 10^4 \text{ J/m}^2$, respectively, are shown in Fig. R13c.

Fig. R13| (a) Schematic of experimental configuration for angle-dependent FMR measurements. (b) Angle-dependent FMR resonance field $H_r(\theta)$ and corresponding Kittel formula fitting (solid lines). (c) K_{eff} versus electric-field curve.

We have added these results as Fig. 2c&d in the main text and Supplementary Fig. S9 in the Supplementary Information of the revised manuscript.

3. While the authors sell this method on the potential of low power control of skyrmions, they leave out discussion on an important issue with ferroelectric material based devices: the ferroelectric polarization fatigue which limits the stability and durability of the devices. Data that shows the repeatability and reversibility of this device/method should be provided to prove that this method is feasible in practical devices.

Response: In our devices, the mechanism of low power control of skyrmions is strain-mediated magnetoelectric coupling. Therefore, the stability and durability of the devices is good if the strain has an excellent endurance. We performed strain measurements using a strain gauge as shown in Fig. R14a in our previous study (Nat. Commun. 10, 243 (2019)). The strain measurement under 8 kV/cm and -1.6 kV/cm electric-field pulses for more than 20000 cycles is shown in Fig. R14b. The statistics of strain of $\pm 0 \text{ kV/cm}$ is shown in Fig. R14c. It can be seen that the strain of PMN-PT is stable and endurance, which suggesting that the devices based on PMN-PT has a good endurance. We also measured the magnetization of the sample switched by pulsed electric fields for thousands of times, and the magnetizations at $\pm 0 \text{ kV/cm}$ are stable (Phys. Rev. Lett. 108, 137203 (2012)). The data does not show any degradation of device performance, indicating the stability and durability of the devices.

[Redacted]

Fig. R14| a, Schematic of the experimental configuration for strain measurements. **b**, The reversible and stable strains stitched by 8 kV/cm and -1.6 kV/cm electric-field pulses for more than 20000 cycles. **c**, The strain distribution at ± 0 kV/cm in **b**. **d**, The switch between the high/low magnetization states by pulsed electric fields for thousands of times.

We have added some discussion and the references in the main text of the revised manuscript (page 7).

4. Because of the above raised concerns, I hesitate to recommend this paper for publication in Nature Communications without a major revision based on extended data.

Response: According to the reviewers' comments and suggestions, we have made a major revision based on extended data.

As a minor note not related to the science in this report, I strongly suggest editing help from someone with full professional proficiency in English.

Response: The revised manuscript was proofread by a professor in the USA and it should be improved remarkably.

A summary of main changes in the revised manuscript

According to the suggestions and comments of the reviewers, we have made a major revision to our paper. The main changes are as follows:

1. According to the comment 1 of reviewer #1 that “The authors should try to include some figures (e.g. Fig. 1: Schematic of the sample configuration, Supplementary Fig. 18: A series of skyrmion morphology under different in-plane biaxial tensile strain.) in order to increase the readability of the paper”, we have added Fig. 1a (Schematic of the sample configuration) in the main text of the revised manuscript. We also improved the micromagnetics simulations by considering both DMI and the effect of pinning on the disappearance/reappearance of one skyrmion rather than the tensile strains (previous Supplementary Fig. 18) and the results are shown in the insets of Fig. 5 in the main text of the revised manuscript.
2. According to the comment 2 of reviewer #1 that “It would be nice for the readability and the reproducibility of the paper to include the actual energy functional used having contributions for instance from exchange, anisotropy, demagnetization and Dzyaloshinski-Moriya or from thermal fields if we refer to temperatures differing from 0 K”, we have added energy functions in the part of Methods-Micromagnetic simulation in the main text of the revised manuscript. And we have also added a discussion on the thermal fields as Supplementary Fig. 19 in the Supplementary Information of the revised manuscript.
3. According to the comment 1 of reviewer #2 that “The authors need to perform the DMI extraction experiments using the same electric field hysteresis loops (i.e. from +4kV/cm to -4kV/cm and not from +8kV/cm to -8kV/cm)”, we have performed the DMI experiments using the same electric field hysteresis loop (from +4 kV/cm to -4 kV/cm) and added these results as Fig. 2b in the main text of the revised manuscript and Supplementary Fig. 8 in the Supplementary Information of the revised manuscript.
4. Because of the updated DMI, we have updated micromagnetic simulations and added these results as Fig. 3e-k in the main text of the revised manuscript and Supplementary Fig. 14-20 & 22 in the Supplementary Information of the revised manuscript.
5. According to the comment 3 of reviewer #2, we have added the magnetic images and their corresponding topography images as Supplementary Fig. 13 in the Supplementary Information of the revised manuscript.

6. According to the comment 4 of reviewer #2 that “Have the authors taken the same values of DMI vs electric field also for simulations in Figs. 4 and 5 or do they only assume a magnetoelastic change of anisotropy in these simulations?”, we have performed the simulations with the change of DMI and added these results as Fig.4 & 5 in the main text of the revised manuscript.
7. According to the comment 4 of reviewer #2 that “Can the simulation reproduce the reappearance of a skyrmion in Fig. 5c if the authors assume/disregard change of DMI with strain?”, we have explored this problem in depth, proposed an explanation, and performed the micromagnetic simulation. And the simulation has reproduced the reappearance of a skyrmion. We added these results as Fig. 5 in the main text of the revised manuscript.
8. According to the comment 5 of reviewer #2 that “I also have some concerns regarding the extraction of DMI from BLS. When using the BLS technique to determine DMI, how do the authors exclude possible non-reciprocal effects due to dipolar interactions, which can be very pronounced in multilayers”, we have performed a series of experiments to exclude the possible non-reciprocal effects due to the dipolar interaction, and added these results as Supplementary Fig. 8 in the Supplementary Information of the revised manuscript.
9. According to the comment of reviewer #2 that “The arrangement of figures and panels is not consistent with the order of their discussion in the text, which makes the manuscript unnecessarily hard to follow”, we have adjusted the arrangement of Fig.2 & 3 to make them consistent with the order of their discussion in the main text of the revised manuscript.
10. According to the comment 1 of reviewer #3, we performed asymmetric domain expansion under an in-plane magnetic field by L-TEM and also added the discussion on skyrmion chirality as Supplementary Fig. 21 in the Supplementary Information of the revised manuscript.
11. According to the comment 2 of reviewer #3, we performed angle-dependent ferromagnetic resonance measurements and added the discussion on electric-field modulation of magnetic anisotropy as Fig. 2c-d in the main text of the revised manuscript and as Supplementary Fig. 9 in the Supplementary Information of the revised manuscript.
12. We have further edited the manuscript to improve readability and corrected some grammatical errors.
13. We have added Fig. 1a, 2c-d in the main text and Fig. S8, S9, S12, S19, S21 in the Supplementary Information of the revised manuscript. We have updated Fig. 2b,

3e-k, 4, 5 in the main text and Fig. S7, S14-S18, S20, S22 in the Supplementary Information of the revised manuscript.

REVIEWERS' COMMENTS

Reviewer #1

Editor note: Rev#1 supports publication.

Reviewer #2 (Remarks to the Author):

The authors replied in detail to the comments of all reviewers and made major revisions to the manuscript. The revised manuscript is much improved. The new measurement data (Fig. 2) is now consistent with the rest of the manuscript. With the additional data added to the SI, the authors could convince me that their extraction of DMI by BLS is indeed valid and not affected by dipolar effects. The authors have also adjusted their micromagnetic simulations to reflect the newly measured parameters and can now demonstrate good agreement with their experiments.

With the additional data and revised manuscript I am now convinced that the authors provide sufficient experimental evidence for a strain-induced change of DMI (and anisotropy, as well known). By this, the authors have resolved my main concern. Together with the revised micromagnetic simulations, they have also provided sufficient evidence for the claimed electric field-control of skyrmions.

The authors have furthermore also addressed and resolved all of my minor comments. The manuscript topic remains highly relevant, novel and interesting to a broad audience. The revised manuscript is much clearer and easier to follow.

Because of this, I now recommend publication in Nature Communications and have no further questions or comments.

Reviewer #3 (Remarks to the Author):

I believe the authors provide necessary revisions and satisfactory replies, and thus recommend publication in Nature Communications.

One minor comment related to the small DMI value in this system is that when the films are made into multilayer structures, the dipolar interaction can become significant so that it can act together with the DMI to induce a skyrmion/stripe magnetic domain phase. In other words, the dipolar interaction along with the DMI stabilizes the overall domain phase, while the small yet non-zero DMI induces (Neel-type) chirality in the domain walls. I think such a comment might convince the readers that a Neel-type skyrmion phase can be stabilized even with the small DMI measured in this study.

Response to the reviewers' comments

Reviewer #1 (Remarks to the Author):

Editor note: Rev#1 supports publication.

Response: We are delighted that our reply answered the reviewer's concerns and appreciate the reviewer's support.

Reviewer #2 (Remarks to the Author):

The authors replied in detail to the comments of all reviewers and made major revisions to the manuscript. The revised manuscript is much improved. The new measurement data (Fig. 2) is now consistent with the rest of the manuscript. With the additional data added to the SI, the authors could convince me that their extraction of DMI by BLS is indeed valid and not affected by dipolar effects. The authors have also adjusted their micromagnetic simulations to reflect the newly measured parameters and can now demonstrate good agreement with their experiments.

With the additional data and revised manuscript I am now convinced that the authors provide sufficient experimental evidence for a strain-induced change of DMI (and anisotropy, as well known). By this, the authors have resolved my main concern. Together with the revised micromagnetic simulations, they have also provided sufficient evidence for the claimed electric field-control of skyrmions.

The authors have furthermore also addressed and resolved all of my minor comments. The manuscript topic remains highly relevant, novel and interesting to a broad audience. The revised manuscript is much clearer and easier to follow.

Because of this, I now recommend publication in Nature Communications and have no further questions or comments.

Response: We are delighted that our reply answered the reviewer's concerns and appreciate the reviewer's recommending publication of our paper in Nature Communications.

Reviewer #3 (Remarks to the Author):

I believe the authors provide necessary revisions and satisfactory replies, and thus recommend publication in Nature Communications.

Response: We are delighted that our reply answered the reviewer's concerns and thank the reviewer's recommending publication of our paper in Nature Communications.

One minor comment related to the small DMI value in this system is that when the films are made into multilayer structures, the dipolar interaction can become significant so that it can act together with the DMI to induce a skyrmion/stripe magnetic domain phase. In other words, the dipolar interaction along with the DMI stabilizes the overall domain phase, while the small yet non-zero DMI induces (Neel-type) chirality in the domain walls. I think such a comment might convince

the readers that a Neel-type skyrmion phase can be stabilized even with the small DMI measured in this study.

Response: Thank the referee for this comment, which is helpful for the readers to have a better understanding of our work. We have added a statement in the revised main text (page 7) as follows.

“The DMI value is a little bit small. However for magnetic multilayer structures, the dipolar interaction can become significant. It has been shown that both the dipolar interaction and the DMI are important for skyrmions in magnetic multilayer structures [58].”

58. Li, W. et al. Anatomy of Skyrmionic Textures in Magnetic Multilayers. *Adv. Mater.* **31**, 1807683 (2019).